# GAT: Generative Adversarial Training for Adversarial Example Detection and Robust Classification

**Xuwang Yin**
Department of Electrical and Computer Engineering
University of Virginia
xy4cm@virginia.edu

**Soheil Kolouri**
Information and Systems Sciences Laboratory
HRL Laboratories, LLC.
skolouri@hrl.com

**Gustavo K. Rohde**
Department of Electrical and Computer Engineering
University of Virginia
gustavo@virginia.edu

## Abstract

The vulnerabilities of deep neural networks against adversarial examples have become a significant concern for deploying these models in sensitive domains. Devising a definitive defense against such attacks is proven to be challenging, and the methods relying on detecting adversarial samples are only valid when the attacker is oblivious to the detection mechanism. In this paper we propose a principled adversarial example detection method that can withstand norm-constrained white-box attacks. Inspired by one-versus-the-rest classification, in a K class classification problem, we train K binary classifiers where the i-th binary classifier is used to distinguish between clean data of class i and adversarially perturbed samples of other classes. At test time, we first use a trained classifier to get the predicted label (say k) of the input, and then use the k-th binary classifier to determine whether the input is a clean sample (of class k) or an adversarially perturbed example (of other classes). We further devise a generative approach to detecting/classifying adversarial examples by interpreting each binary classifier as an unnormalized density model of the class-conditional data. We provide comprehensive evaluation of the above adversarial example detection/classification methods, and demonstrate their competitive performances and compelling properties. Code is available at https://github.com/xuwangyin/GAT-Generative-Adversarial-Training[1].

## 1 Introduction

Deep neural networks have become the staple of modern machine learning pipelines, achieving state-of-the-art performance on extremely difficult tasks in various applications such as computer vision (He et al., 2016), speech recognition (Amodei et al., 2016), machine translation (Vaswani et al., 2017), robotics (Levine et al., 2016), and biomedical image analysis (Shen et al., 2017). Despite their outstanding performance, these networks are shown to be vulnerable against various types of adversarial attacks, including evasion attacks (aka, inference or perturbation attacks) (Szegedy et al., 2013; Goodfellow et al., 2014; Carlini & Wagner, 2017b; Su et al., 2019) and poisoning attacks (Liu et al., 2017; Shafahi et al., 2018). These vulnerabilities in deep neural networks hinder their deployment in sensitive domains including, but not limited to, health care, finances, autonomous driving, and defense-related applications and have become a major security concern.

Due to the mentioned vulnerabilities, there has been a recent surge toward designing defense mechanisms against adversarial attacks (Gu & Rigazio, 2014; Jin et al., 2015; Papernot et al., 2016b;

---

[1] A thorough evaluation of the proposed method is available at Tramer et al. (2020).

Bastani et al., 2016; Madry et al., 2017; Sinha et al., 2018), which has in turn motivated the design of stronger attacks that defeat the proposed defenses (Goodfellow et al., 2014; Kurakin et al., 2016b;a; Carlini & Wagner, 2017b; Xiao et al., 2018; Athalye et al., 2018; Chen et al., 2018; He et al., 2018). Besides, the proposed defenses have been shown to be limited and often not effective and easy to overcome (Athalye et al., 2018). Alternatively, a large body of work has focused on detection of adversarial examples (Bhagoji et al., 2017; Feinman et al., 2017; Gong et al., 2017; Grosse et al., 2017; Metzen et al., 2017; Hendrycks & Gimpel, 2017; Li & Li, 2017; Xu et al., 2017; Pang et al., 2018; Roth et al., 2019; Bahat et al., 2019; Ma et al., 2018; Zheng & Hong, 2018; Tian et al., 2018). While training robust classifiers focuses on maintaining performance in presence of adversarial examples, adversarial detection only cares for detecting such examples.

The majority of the current detection mechanisms focus on *non-adaptive* threats, for which the attacks are not specifically tuned/tailored to bypass the detection mechanism, and the attacker is oblivious to the detection mechanism. In fact, Carlini & Wagner (2017a) and Athalye et al. (2018) showed that the detection methods presented in (Bhagoji et al., 2017; Feinman et al., 2017; Gong et al., 2017; Grosse et al., 2017; Metzen et al., 2017; Hendrycks & Gimpel, 2017; Li & Li, 2017; Ma et al., 2018), are significantly less effective than their claims under *adaptive* attacks. Overall, current solutions are mostly heuristic approaches that cannot provide performance guarantees.

In this paper we propose a detection mechanism that can withstand adaptive attacks. The idea is to partition the input space into subspaces based on the original classifier's decision boundary, and then perform clean/adversarial example classification the subspaces. The binary classifier in each subspace is trained to distinguish in-class samples from adversarially perturbed samples of other classes. At inference time, we first use the original classifier to get an input sample's predicted label $\hat{k}$, and then use the $\hat{k}$-th binary classifier to identify whether the input is a clean sample (of class $\hat{k}$) or an adversarially perturbed sample (of other classes). Fig. 1 provides a schematic illustration of the proposed approach.

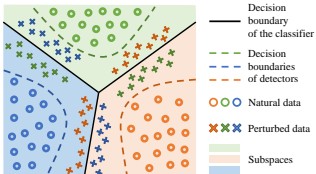

Figure 1: A conceptual visualization of the proposed adversarial example detection mechanism.

Our specific contributions are: (1) We develop a principled adversarial example detection method that can withstand adaptive attacks. Empirically, our best models improve previous state-of-the-art mean $L_2$ distortion from 3.68 to 5.65 on MNIST dataset, and from 1.1 to 1.5 on CIFAR10 dataset. (2) We study powerful and versatile generative classification models derived from our detection framework and demonstrate their competitive performances over discriminative robust classifiers. While discriminative robust classifiers are vulnerable to rubbish examples, inputs that have confident predictions under our models have interpretable features.

## 2 RELATED WORKS

**Adversarial attacks** Since the pioneering work of Szegedy et al. (2013), a large body of work has focused on designing algorithms that achieve successful attacks on neural networks (Goodfellow et al., 2014; Moosavi-Dezfooli et al., 2016; Kurakin et al., 2016b; Chen et al., 2018; Papernot et al., 2016a; Carlini & Wagner, 2017b). More recently, iterative projected gradient descent (PGD), has been empirically identified as the most effective approach for performing norm constrained attacks, and the attack reasonably approximates the optimal attack (Madry et al., 2017).

**Adversarial example detection** The majority of the methods developed for detecting adversarial attacks are based on the following core idea: given a trained $K$-class classifier, $f : \mathbb{R}^d \to \{1...K\}$, and its corresponding clean training samples, $\mathcal{D} = \{x_i \in \mathbb{R}^d\}_{i=1}^{N}$, generate a set of adversarially attacked samples $\mathcal{D}' = \{x_j' \in \mathbb{R}^d\}_{j=1}^{M}$, and devise a mechanism to discriminate $\mathcal{D}$ from $\mathcal{D}'$. For instance, Gong et al. (2017) use this exact idea and learn a binary classifier to distinguish the clean and adversarially perturbed sets. Similarly, Grosse et al. (2017) append a new "attacked" class to the classifier, $f$, and re-train a secured network that classifies clean images, $x \in \mathcal{D}$, into the $K$ classes and all attacked images, $x' \in \mathcal{D}'$, to the $(K + 1)$-th class. In contrast to Gong et al. (2017); Grosse et al. (2017), which aim at detecting adversarial examples directly from the image content, Metzen et al. (2017) trained a binary classifier that receives as input the intermediate layer features extracted from the classifier network $f$, and distinguished $\mathcal{D}$ from $\mathcal{D}'$ based on such input features. More

importantly, Metzen et al. (2017) considered the so-called case of *adaptive/dynamic* adversary and proposed to harden the detector against such attacks using a similar adversarial training approach as in Goodfellow et al. (2014). Unfortunately, the mentioned detection methods are significantly less effective under an *adaptive* adversary equipped with a strong attack (Carlini & Wagner, 2017a; Athalye et al., 2018).

# 3 PROPOSED APPROACH TO DETECTING ADVERSARIAL EXAMPLES

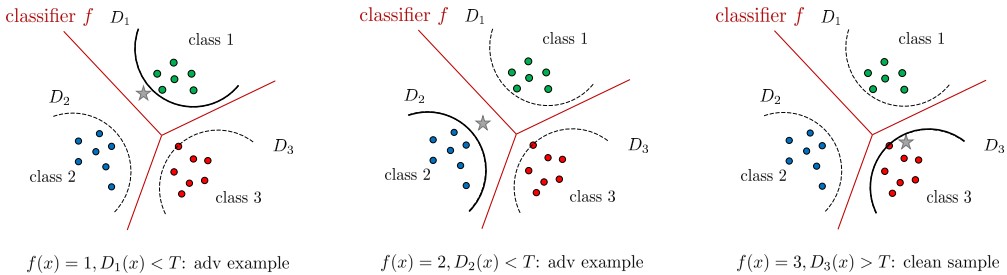

$$f(x) = 1, D_1(x) < T: \text{adv example} \qquad f(x) = 2, D_2(x) < T: \text{adv example} \qquad f(x) = 3, D_3(x) > T: \text{clean sample}$$

Figure 2: A schematic illustration of the proposed method for determining whether an input sample $x$ (represented by the gray star) is an adversarial example. The first figure shows the case where $x$ is predicted by $f$ as class 1 and then $x$ is identified as an adversarial example by $D_1$. The following two figures shows the other two cases where $x$ is respectively predicted as class 2 and class 3 and then $D_2$ and $D_3$ is respectively used to predict whether $x$ is an adversarial example.

The proposed approach to detecting adversarial examples is based on the following simple idea. Assume there is an input sample $x$, and it is predicted as $\hat{k}$ by the classifier $f$, then $x$ is either a true sample of class $\hat{k}$ (assuming no misclassification) or an adversarially perturbed sample of other classes. To determine which is the case we can use a binary classifier that is specifically trained to distinguish between clean samples of class $\hat{k}$ and adversarially perturbed samples of other classes. Because $\hat{k}$ can be any one of the $K$ class, we need to train a total of $K$ binary classifier in order to have a complete solution. Fig. 2 provides a schematic illustration of the above detection idea. We next provide a mathematical justification and more details about how to train the detection model.

In a $K(K \geq 2)$ class classification problem, given a dataset of clean samples $\mathcal{D} = \{x_i\}_{i=1}^N, x_i \in \mathbb{R}^d$, along with labels $\{y_i\}_{i=1}^N, y_i \in \{1, ..., K\}$, let $f : \mathbb{R}^d \to \{1, ..., K\}$ be a classifier on $\mathcal{D}$, and $\mathcal{D}'$ be a set of $p$-norm bounded adversarial examples computed from $\mathcal{D}$: $\mathcal{D}' = \{x + \delta : f(x + \delta) \neq y, f(x) = y, x \in \mathcal{D}, \delta \in \mathcal{S}\}, \mathcal{S} = \{\delta \in \mathbb{R}^d \mid \|\delta\|_p \leq \epsilon\}$. We use $\mathcal{D}_k^f = \{x : f(x) = k, x \in \mathcal{D}\}$ and $\mathcal{D}'_k^f = \{x : f(x) = k, x \in \mathcal{D}'\}$ to respectively denote the clean samples and adversarial examples which are predicted by $f$ as class $k$. Let $\mathcal{H} = \{d_k\}_{k=1}^K$, where $d_k : \mathbb{R}^d \to [0, 1]$ is a binary classifier trained to distinguish between samples from $\mathcal{D}_k^f$ (assigned as class 1) and samples from $\mathcal{D}'_k^f$ (assigned as class 0). We use $d_k(x)$ to model $p(x \in \mathcal{D}_k^f | x)$ and predict $x \in \mathcal{D}_k^f$ when $d_k(x) > \frac{1}{2}$ and $x \in \mathcal{D}'_k^f$ when $d_k(x) \leq \frac{1}{2}$. Consider the following procedure to determine whether a sample $x$ is an adversarial example (i.e., whether it comes from $\mathcal{D}$ or $\mathcal{D}'$):

> *First obtain the estimated class label $\hat{k} = f(x)$, then use the $\hat{k}$-th binary classifier to predict: if $d_{\hat{k}}(x) >= \frac{1}{2}$ then categorize $x$ as a clean sample, otherwise categorize it as an adversarial example.*

This algorithm can be viewed as a binary classifier, and its accuracy is given by

$$\frac{\sum_{k=1}^K |\{x : d_k(x) > \frac{1}{2}, x \in \mathcal{D}_k^f\}| + |\{x : d_k(x) \leq \frac{1}{2}, x \in \mathcal{D}'_k^f\}|}{|\mathcal{D}| + |\mathcal{D}'|}. \tag{1}$$

Crucially, because the errors of individual binary classifiers are independent, maximizing Eq. (1) is equivalent to optimizing the performances of individual binary classifiers. $d_k$ solves the binary classification problem of distinguishing between samples from $\mathcal{D}_k^f$ and samples from $\mathcal{D}'_k^f$, and therefore

can be trained with a binary classification objective:

$$\theta_k^* = \arg\min_{\theta_k} \mathbb{E}_{x \sim \mathcal{D}'^f_k}\Big[L(d_k(x;\theta_k),0)\Big] + \mathbb{E}_{x \sim \mathcal{D}^f_k}\Big[L(d_k(x;\theta_k),1)\Big], \tag{2}$$

where $L$ is a loss function that measures the discrepancy between $d_k$'s output and the supplied label (e.g., the negative log likelihood loss). In order to harden $d_k$ against adaptive attacks, we follow Madry et al. (2017) and incorporate the adversary into the training objective:

$$\min_{\theta_k} \rho(\theta_k), \quad \text{where} \quad \rho(\theta_k) = \mathbb{E}_{x \sim \mathcal{D}^f_{\setminus k}}\Big[\max_{\delta \in \mathcal{S}, f(x+\delta)=k} L(d_k(x+\delta;\theta_k),0)\Big] + \mathbb{E}_{x \sim \mathcal{D}^f_k}\Big[L(d_k(x;\theta_k),1)\Big], \tag{3}$$

where $\mathcal{D}^f_{\setminus k} = \{x : f(x) \neq k, y \neq k, x \in \mathcal{D}\}$, and we assume that $\epsilon$ is large enough such that $\forall x \in \mathcal{D}^f_{\setminus k}$, $\exists \delta \in \mathcal{S}$, s.t. $f(x+\delta) = k$.

The equality constraint $f(x+\delta) = k$ in Eq. (3) complicates the inner maximization. We observe that by dropping this constrain we have the following upper bound of the first loss term:

$$\max_{\delta \in \mathcal{S}, f(x+\delta)=k} L(d_k(x+\delta;\theta_k),0) \leq \max_{\delta \in \mathcal{S}} L(d_k(x+\delta;\theta_k),0).$$

Because we are minimizing $L(d_k(x+\delta;\theta_k),0)$, we can instead minimizing this upper bound, which gives us the unconstrained objective

$$\rho(\theta_k) = \mathbb{E}_{x \sim \mathcal{D}^f_{\setminus k}}\Big[\max_{\delta \in \mathcal{S}} L(d_k(x+\delta;\theta_k),0)\Big] + \mathbb{E}_{x \sim \mathcal{D}^f_k}\Big[L(d_k(x;\theta_k),1)\Big]. \tag{4}$$

We can further simply this objective by using the fact that when $\mathcal{D}$ is used as the training set, $f$ can overfit on $\mathcal{D}$ such that $\mathcal{D}_{\setminus k} = \{x_i : y_i \neq k\}$ and $\mathcal{D}_k$ are respectively good approximations of $\mathcal{D}^f_{\setminus k}$ and $\mathcal{D}^f_k$:

$$\min_{\theta_k} \rho(\theta_k), \quad \text{where} \quad \rho(\theta_k) = \mathbb{E}_{x \sim \mathcal{D}_{\setminus k}}\Big[\max_{\delta \in \mathcal{S}} L(d_k(x+\delta;\theta_k),0)\Big] + \mathbb{E}_{x \sim \mathcal{D}_k}\Big[L(d_k(x;\theta_k),1)\Big]. \tag{5}$$

In words, each binary classifier is trained using clean samples of a particular class and adversarial examples (with respect to $d_k$) created from samples of other class. The inner maximization is solved using the PGD attack (Madry et al., 2017). We use the negative log likelihood loss as $L$ and minimize it using gradient-based optimization methods.

From a classification point of view, we can reformulate the above detection algorithm as a *classifier* that has a rejection option: given input $x$ and its prediction label $\hat{k} = f(x)$, if $d_{\hat{k}}(x) < T$, then $x$ is rejected, otherwise it's classified as $\hat{k}$. We will refer to this classification model as an *integrated classifier*.

### 3.1 A Generative approach to adversarial example detection/classification

The proposed approach makes use of a trained classifier $f$ to get the predicted label, but $f$ is not strictly necessary: we can use $\mathcal{H} = \{d_k\}_{k=1}^K$ in place of $f$ to do classification.

We can interpret $\mathcal{H}$ as an *one-versus-the-rest* (OVR) classifier. In a $K$ class classification problem, a OVR classifier consists of $K$ binary classifiers, with each one trained to solve a two-class problem of separating samples in a particular class from samples not in that class. $\mathcal{H}$ differs from a traditional OVR classifier in that $d_k$ is trained to distinguish between samples in class $k$ and *adversarially perturbed samples* of other classes, but because the loss on adversarial inputs is an upper bound of the loss on clean samples, the binary classifier should also be able to separate samples of class $k$ from *clean samples* of other classes. When $\mathcal{H}$ is viewed as an OVR classifier, the classification rule is

$$H(x) = \arg\max_{k} d_k(x). \tag{6}$$

We can also interpret $\mathcal{H}$ as a *generative classifier*. Our experiments show that $d_k$ has a strong generative property: performing adversarial attacks on $d_k$ causes visual features of class $k$ to appear in the attacked data (in some cases, the attacked data become a valid instance of class $k$). Although

a similar phenomenon is observed in standard adversarial training (Tsipras et al., 2018; Engstrom et al., 2019; Santurkar et al., 2019), the generative property of our model seems to be much stronger than that of a softmax adversarially robust classier (Fig. 4, Fig. 6, and Fig. 7). These results motivate us to reinterpret $d_k$ as an unnormalized density model (i.e., an energy-based model (LeCun et al., 2006)) of the class-$k$ data. This interpretation allows us to obtain the class-conditional probability of an input by:

$$p(x|k) = \frac{\exp(-E_k(x))}{Z_k}, \tag{7}$$

where $E_k(x) = -z_{d_k}(x)$, with $z_{d_k}$ being the logit output of $d_k$, and

$$Z_k = \int \exp(-E_k(x))dx \tag{8}$$

is an intractable normalizing constant known as the partition function. We can then apply the Bayes classification rule to obtain a *generative classifier*:

$$H(x) = \arg\max_k p(k|x) = \arg\max_k \frac{p(x|k)p(k)}{p(x)} = \arg\max_k z_{d_k}(x), \tag{9}$$

where we have assumed all partition functions $Z_k, k = 1, ..., K$ and class priors $p(k), k = 1, ..., K$ to be equal. Because we explicitly model $p(x, k)$, we can use this quantity to reject low probability inputs which can be any samples that do not belong to class $k$. In this work we focus on the scenario where low probability inputs are adversarially perturbed samples of other classes and the rejected samples are considered as adversarial examples. Because $d_k(x)$ is computed by applying the logistic sigmoid function to $z_{d_k}(x)$, and the logistic sigmoid function is a monotonically increasing function of its argument, the generative classifier (Eq. (9)) is equivalent to the OVR classifier (Eq. (6)).

In the following sections, we will use *integrated detection* to refer to the original detection approach where we make use an extra classifier $f$, and *generative detection* to refer to this alternative approach where we first use the generative classifier Eq. (9) to get the predicted label $\hat{k}$ of an input $x$, and then use $d_{\hat{k}}$ to determine whether $x$ is adversarial input.

## 4 EVALUATION METHODOLOGY

### 4.1 ROBUSTNESS TEST

We first validate the robustness of individual binary classifiers by following the standard methodology for robustness testing: we train the binary classifier with PGD attack configured with a particular combination of step-size and number of steps, and then test the binary classifier's performance under PGD attacks configured with different combinations of step-sizes and number of steps. We use AUC (area under the ROC Curve) as the detection performance metric. AUC is an aggregated measurement of detection performance across a range of thresholds, and can be interpreted as the probability that the binary classifier assigns a higher score to a random positive sample than to a random negative example. For a given $d_k$, the AUC is computed on the set $\{(x, 0) : x \in \mathcal{D}'^f_{\backslash k}\} \cup \{(x, 1) : x \in \mathcal{D}^f_k\}$, where $\mathcal{D}'^f_{\backslash k} = \{x + \arg\max_\delta L(d_k(x + \delta; \theta_k), 0) : x \in \mathcal{D}^f_{\backslash k}\}$.

### 4.2 ADVERSARIAL EXAMPLE DETECTION PERFORMANCE

Having validated the robustness of individual binary classifier, we evaluate the overall performance of the proposed approach to detecting adversarial examples. According to the detection algorithm, we first obtain the predicted label $\hat{k} = f(x)$, and then use the $\hat{k}$-th binary classifier's logit output to predict: if $z_{d_{\hat{k}}}(x) \geq T$, then $x$ is a clean sample, otherwise it is an adversarially perturbed sample.

We use $\mathcal{D} = \{(x_i, y_i)\}_{i=1}^N$ to denote the test set that contains clean samples, and $\mathcal{D}' = \{(x_i + \delta_i, y_i)\}_{i=1}^N$ to denote the corresponding perturbed test set. For a given $T$, we compute the true positive rate (TPR) on $\mathcal{D}$ and false positive rate (FPR) on $\mathcal{D}'$ (here, clean samples are in the positive class). These two metrics are respectively defined as

$$\text{TPR} = \frac{1}{|\mathcal{D}|} |\{x : z_{d_{\hat{k}}}(x) \geq T, k = f(x), (x, y) \in \mathcal{D}\}|, \tag{10}$$

and

$$\text{FPR} = \frac{1}{|\mathcal{D}'|} |\{x : z_{d_{\hat{k}}}(x) \geq T, k = f(x), f(x) \neq y, (x, y) \in \mathcal{D}'\}|. \tag{11}$$

We observe that for the norm ball constraint we considered in the experiments, not all perturbed samples can cause misclassification on $f$, so we use $f(x) \neq y$ in the FPR definition to constrain that *only adversarial inputs that actually cause misclassification* can be counted as false positives.

Given a clean sample $x$ and its groundtruth label $y$, we consider three approaches to creating the corresponding adversarial example $x'$. Here we will focus on untargeted attacks.

**Classifier attack** This attack corresponds to the scenario where the adversary is oblivious to the detection mechanism. Inspired by the CW attack (Carlini & Wagner, 2017b), the adversarial example $x'$ is computed by minimizing,

$$L(x') = z_f(x')_y - \max_{i \neq y} z_f(x')_i, \tag{12}$$

where $z_f(x')$ is the classifier's logit outputs.

**Detector attack** In this scenario adversarial examples are produced by attacking *only* the detector. We first construct a detection function $H$ by aggregating the logit outputs of individual binary classifiers:

$$z_H(x)_i = z_{d_i}(x). \tag{13}$$

The adversarial example $x'$ is then computed by minimizing

$$L(x') = -\max_{i \neq y} z_H(x')_i. \tag{14}$$

According to our detection rule, a low value of a binary classifier's logit output indicates the detection of an adversarial example, and therefore by minimizing the negative of the logit output we make the adversarial input harder to detect. $H$ can also be used with the CW loss Eq. (12) or the cross-entropy loss, but we find the attack based on Eq. (14) to be most effective.

**Combined attack** The combined attack is an adaptive method that considers both the classifier and the detector. We consider two loss functions for the combined attack. The first is based on the adaptive attack of Carlini & Wagner (2017a) which has been shown to be effective against existing detection methods. We first construct a new detection function $H$ with Eq. (13) and then use $H$'s largest logit output $\max_{k \neq y} z_H(x)_k$ (low value of this quantity indicates detection of an adversarial example) and the classifier logit outputs $z_f(x)$ to construct a new classifier $g$:

$$z_g(x)_i = \begin{cases} z_f(x)_i & \text{if } i \leq K, \\ (-\max_{j \neq y} z_H(x)_j + 1) \cdot \max_j z_f(x)_j & \text{if } i = K + 1. \end{cases} \tag{15}$$

The adversarial example $x'$ is then computed by minimizing the loss function

$$L(x') = \max_i z_g(x')_i - \max_{i \neq y} z_f(x')_i. \tag{16}$$

In practice we observe that the optimization of Eq. (16) tends to stuck at the point where $\max_{i \neq y} z_f(x')_i$ keeps changing signs while $\max_{j \neq y} z_H(x)_j$ staying as a large negative number (which indicates detection). In light of the above issues we derive a more effective attack by combining Eq. (12) and Eq. (14):

$$L(x') = \begin{cases} z_f(x')_y - \max_{i \neq y} z_f(x')_i & \text{if } z_f(x')_y \geq \max_{i \neq y} z_f(x')_i, \\ -\max_{i \neq y} z_H(x')_i & \text{else.} \end{cases} \tag{17}$$

In words, if $x'$ is not yet an adversarial example on $f$ (case 1), optimize it for that goal, otherwise optimize it for evading the detection (case 2).

We note that the above three attacks are for the original detection approach (i.e., integrated detection). The *generative detection* approach (Section 3.1) does not make use of $f$ and we use Eq. (14) to create adversarial examples for generative detection.

### 4.3 ROBUST CLASSIFICATION PERFORMANCE

In robust classification, the accuracy of a softmax robust classifier is evaluated on the original test test (the *standard accuracy*) and the adversarially perturbed test set (the *robust accuracy*). Because the generative classifier comes with the reject option, we use slightly different metrics. On the clean test dataset $\mathcal{D} = \{(x_i, y_i)\}_{i=1}^{N}$, we define the *standard accuracy* as the fraction of samples that are correctly classified ($f(x) = y$) and at the same time not rejected ($z_{d_{\hat{k}}}(x) \geq T$):

$$\text{Accuracy} = \frac{1}{N}|\{x : z_{d_k}(x) \geq T, k = f(x), f(x) = y, (x, y) \in \mathcal{D}\}|. \tag{18}$$

In the adversarially perturbed test dataset $\mathcal{D}' = \{(x_i + \delta_i^*, y_i)\}_{i=1}^{N}$, we will consider a data sample as properly handled when it is rejected ($z_{d_{\hat{k}}}(x) < T$), regardless of whether it causes misclassification. In this way, only misclassified ($f(x) \neq y$) and unrejected ($z_{d_{\hat{k}}}(x) \geq T$) samples are counted as errors:

$$\text{Error} = \frac{1}{N}|\{x : z_{d_k}(x) \geq T, k = f(x), f(x) \neq y, (x, y) \in \mathcal{D}'\}|. \tag{19}$$

To compare different classifiers under the same metrics, we compute the error of a softmax robust classifier $g$ on $\mathcal{D}'$ as

$$\text{Error} = \frac{1}{N}|\{x : g(x) \neq y, (x, y) \in \mathcal{D}'\}|. \tag{20}$$

We respectively use Eq. (17) and Eq. (14) to compute the $\mathcal{D}'$ for the integrated classifier and the generative classifier.

## 5 EXPERIMENTS

### 5.1 MNIST

We train four detection models (each consists of ten binary classifiers) by using different combinations of $p$-norm and perturbation limit $\epsilon$ (Table 5). The adversarial examples used for training and validation are computed using PGD attacks of different steps and step sizes (Table 5). At each step of PGD attack we use the Adam optimizer to perform gradient descent, both for $L_2$-based and $L_\infty$-based attacks. Appendix A.1 provides more training details.

**Robustness results** Table 1 and Table 7 show that $d_0$ and $d_1$ are able to withstand PGD attacks configured with different steps and step-size, for both $L_2$-based and $L_\infty$-based attacks. The binary classifiers also exhibit robustness when the attack uses $p$-norm or perturbation limit that are different from those used for training the model (Table 8). The models are also robust when the attacks use multiple random restarts (Table 9).

Table 1: AUC scores of the first two binary classifiers $(d_1, d_2)$ tested with different configurations of PGD attacks. In each step of the PGD attack we use the Adam optimizer to perform gradient descent.

| PGD attack steps, step size | $L_\infty$ $\epsilon = 0.3$ model | | $L_\infty$ $\epsilon = 0.5$ model | | PGD attack steps, step size | $L_2$ $\epsilon = 2.5$ model | | $L_2$ $\epsilon = 5.0$ model | |
| --- | --- | --- | --- | --- | --- | --- | --- | --- | --- |
| | $d_1$ | $d_2$ | $d_1$ | $d_2$ | | $d_1$ | $d_2$ | $d_1$ | $d_2$ |
| 200, 0.01 | 0.99959 | 0.99971 | 0.99830 | 0.99869 | 200, 0.1 | 0.99962 | 0.99968 | 0.99578 | 0.99987 |
| 2000, 0.005 | 0.99958 | 0.99971 | 0.99796 | 0.99861 | 2000, 0.05 | 0.99927 | 0.99900 | 0.99529 | 0.99918 |

Table 2: Mean $L_2$ distortion (higher is better) of perturbed samples when the detection method has 1.0 FPR on the perturbed MNIST test set and 0.95 TPR on the clean MNIST test set.

| Detection method | Mean $L_2$ distortion |
| --- | --- |
| State-of-the-art (Carlini & Wagner, 2017a) | 3.68 |
| Ours (generative detection with $L_\infty$ $\epsilon = 0.3$ binary classifiers) | 4.40 |
| Ours (generative detection with $L_\infty$ $\epsilon = 0.5$ binary classifiers) | 5.65 |

**Detection results** Fig. 3a shows the performances of integrated detection and generative detection under different attacks. Combined attack with Eq. (17) is the most effective attack against integrated

detection, and is much more effective than the combined attack with Eq. (16). Overall, generative detection outperforms integrated detection when they are evaluated under their respective most effective attack. It is also interesting to note that when the adversarial examples are created by attacking only the classifier (Eq. (12)), integrated detection is able to perfectly detect these adversarial examples (see the red curve that overlaps the y-axis).

Given that generative detection is the most effective approach among the proposed approaches, we compare it with state-of-the-art detection methods (Carlini & Wagner, 2017a). Table 2 shows that generative detection outperforms the state-of-the-art method by large margins. Appendix B provides details about how the mean $L_2$ distortions are computed.

**Classification results** Figure 3b shows the standard and robust classification performances of the proposed classifiers and a state-of-the-art softmax robust classifier (Madry et al., 2017). Our models provide the reject option that allows the user to find a balance between standard accuracy and robust error by adjusting the rejection threshold. We observe that a stronger attack ($\epsilon = 0.4$) breaks the softmax robust classifier (as indicated by the right red cross), while the generative classifier still exhibits robustness, even though both models are trained with the $L_\infty$ $\epsilon = 0.3$ constraint.

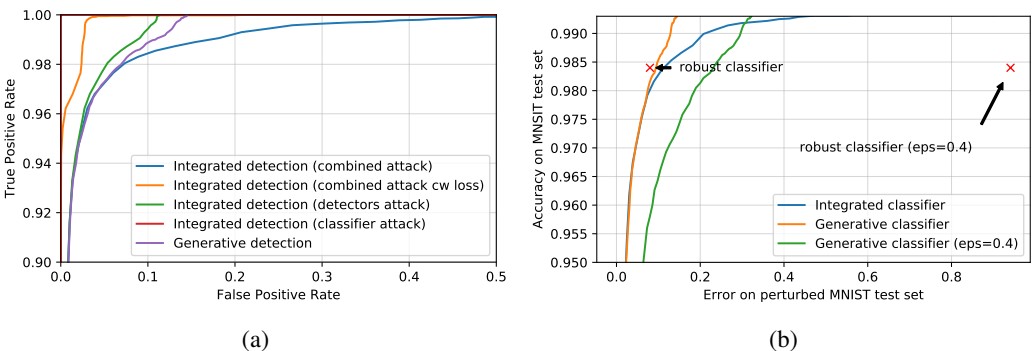

(a)                  (b)

Figure 3: (a) Performances of integrated detection and generative detection under $L_\infty$ $\epsilon = 0.3$ constrained attacks. (b) Performances of the integrated classifier and generative classifier under $L_\infty$ $\epsilon = 0.3$ constrained and $L_\infty$ $\epsilon = 0.4$ constrained attacks. The performances of the softmax robust classifier (Madry et al., 2017) (accuracy 0.984, error 0.08 at $\epsilon = 0.3$, and accuracy 0.984, error 0.941 at $\epsilon = 0.4$) are marked with red crosses. PGD attack steps 100, step size 0.01.

Figure 4 shows perturbed samples produced by performing targeted attacks against the generative classifier and softmax robust classifier. The generative classifier's perturbation samples have distinguishable visible features of the target class, indicating that individual binary classifiers have learned the class conditional distributions, and the perturbations have to change the semantics for a successful attack. In contrast, perturbations introduced by attacking the softmax robust classifier are not interpretable, even though they can cause high logit output of the target classes (see Figure 9 for the logit outputs distribution).

## 5.2 CIFAR10

On CIFAR10 we train a single detection model that consists of ten binary classifiers using $L_\infty$ $\epsilon = 8$ constrain PGD attack of steps 40 and step size 0.5 (note that the scale of $\epsilon$ and step size here is 0-255, as opposed to 0-1 as in the case of MNIST). The softmax robust classifier (Madry et al., 2017) that we compare with is also trained with $L_\infty$ $\epsilon = 8$ constraint but with a different step size (Appendix C.2.2 provides a discussion on the effects of step size). Appendix A.2 provides the training details.

**Robustness results** Table 3 shows that $d_1$ and $d_2$ can withstand PGD attacks configured with different steps and step-size. In Appendix C.2.1 we report random restart test results, cross-norm and cross-perturbation test results, and robustness test result for $L_2$ based models.

**Detection results** Consistent with the MNIST result, Fig. 5 shows that combined attack with Eq. (17) is the most effective attack against integrated detection, and generative detection similarly outperforms integrated detection. Table 4 shows that generative detection outperforms the state-of-the-art adversarial detection method.

Natural samples      Perturbed samples (generative classifier)      Perturbed samples (robust classifier)

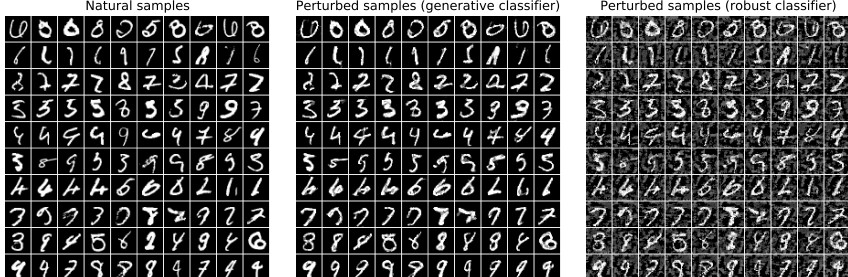

Figure 4: Clean samples and corresponding perturbed samples produced by performing a targeted attack against the generative classifier and robust classifier (Madry et al., 2017). Targets from top row to bottom row are digit class from 0 to 9. We perform the targeted attack by maximizing the logit output of the targeted class, using $L_\infty$ $\epsilon = 0.4$ constrained PGD attack of steps 100 and step size 0.01. Both classifiers are trained with $L_\infty$ $\epsilon = 0.3$ constraint.

Table 3: AUC scores of the first two CIFAR10 $L_\infty$ $\epsilon = 8$ binary classifiers $(d_1, d_2)$ under $L_\infty$ $\epsilon = 8$ constrained PGD attacks of different steps and step-size.

| PGD attack steps, step-size | $d_1$ | $d_2$ |
|---|---|---|
| 20, 2.0 | 0.9224 | 0.9533 |
| 40, 0.5 | 0.9234 | 0.9553 |
| 200, 0.1 | 0.9231 | 0.9550 |
| 200, 0.5 | 0.9205 | 0.9504 |
| 500, 0.5 | 0.9203 | 0.9500 |

Table 4: CIFAR10 mean $L_2$ distortion (higher is better) of perturbed samples when the detection method has 1.0 FPR on perturbed set and 0.95 TPR on the clean set. Appendix B provides details about how the mean $L_2$ distances are computed.

| Detection method | Mean $L_2$ distortion (0-1 scale) |
|---|---|
| State-of-the-art (Carlini & Wagner, 2017a) | 1.1 |
| Ours (generative detection) | 1.5 |

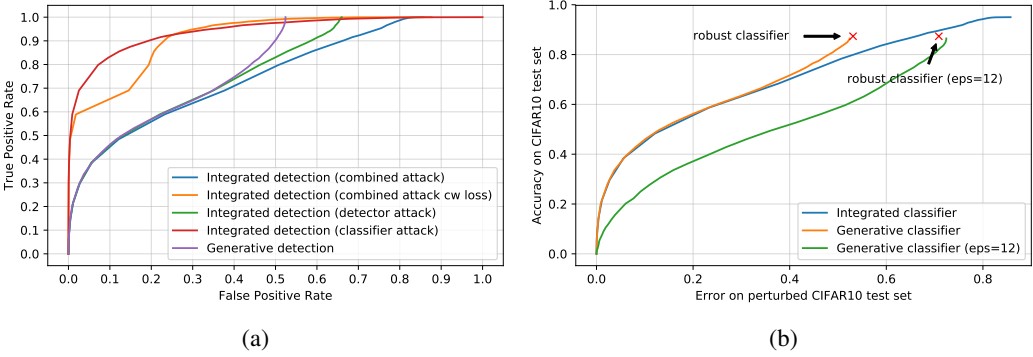

(a)             (b)

Figure 5: (a) Performances of generative detection and integrated detection under $L_\infty$ $\epsilon = 8$ attack. (b) Performances of integrated classifier (discussed in Section 4.3) and generative classifier under $L_\infty$ $\epsilon = 8$ constrained and $L_\infty$ $\epsilon = 12$ constrained attacks. The performances of the robust classifier (Madry et al., 2017) (accuracy 0.8735, error 0.5311 at $\epsilon = 8$, and accuracy 0.8735, error 0.7087 at $\epsilon = 12$) are annotated. PGD attack step size 2.0, steps 20 for $\epsilon = 8$, and 30 for $\epsilon = 12$.

**Classification results** Contrary to MNIST's result, we did not observe a dramatic decrease in the softmax robust classifier's performance when we increase the perturbation limit to $\epsilon = 12$ (Fig. 5b). Integrated classification can reach the standard accuracy of a regular classifier, but at the cost of significantly increased error on the perturbed set. Fig. 6 shows some perturbed samples produced

by attacking the generative classifier and robust classifier. While these two classifiers have similar errors on the perturbed set, samples produced by attacking the generative classifier have more visible features of the attacked classes, suggesting that the adversary needs to change more semantic to cause the same error.

Fig. 7 and Fig. 11 demonstrate that unrecognizable images are able to cause high logit outputs of the softmax robust classifier. This phenomenon highlights a major defect of the softmax robust classifier: they can be easily fooled by unrecognizable inputs (Nguyen et al., 2015; Goodfellow et al., 2014; Schott et al., 2018). In contrast, samples that cause high logit outputs of the generative classifier all have clear semantic meaning. In Figure 14 we present image synthesis results using $L_\infty \epsilon = 16$ constrained detectors. In Appendix D we provide Gaussian noise attack results and a discussion about the interpretability of the generative classification approach.

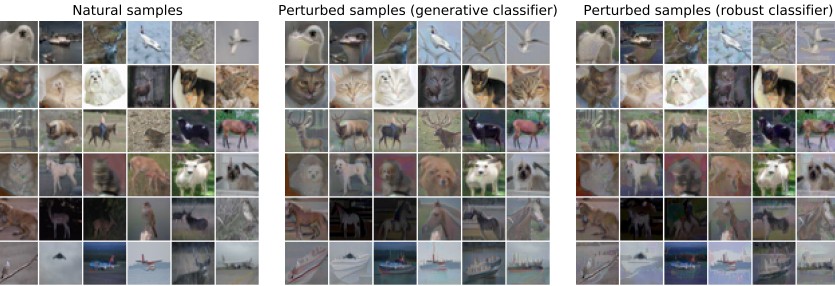

Figure 6: Clean samples and corresponding perturbed samples by performing targeted attack against the generative classifier and robust classifier (Madry et al., 2017). The targeted attack is performed by maximizing the logit output of the targeted class. We use $L_\infty \epsilon = 12$ constrained PGD attack of steps 30 and step size 2.0 to produce these samples.

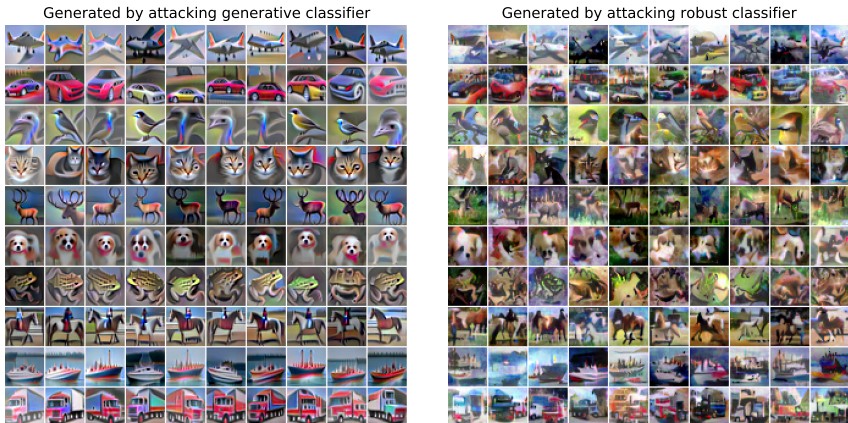

Figure 7: Images generated from class conditional Gaussian noise by performing targeted attack against the generative classifier and robust classifier. We use PGD attack of steps 60 and step size $0.5 \times 255$ to perform $L_2 \epsilon = 30 \times 255$ constrained attack (same as Santurkar et al. (2019). The Gaussian noise inputs from which these two plots are generated are the same. Samples not selected.

## 6  CONCLUSION

We studied the problem of adversarial example detection under the robust optimization framework and proposed a novel detection method that can withstand adaptive attacks. Our formulation leads to a new generative modeling technique which we called generative adversarial training (GAT). GAT's capability to learn class conditional distributions further gives rise to generative detection/classification approaches that show competitive performance and improved interpretability.

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

## A    TRAINING DETAILS

### A.1    MNIST TRAINING

We use 50K samples from the original training set for training and the remaining 10K samples for validation, and report the test performance based on the checkpoint which has the best validation performance. All binary classifiers are trained for 100 epochs, where in each iteration we sample 32 in-class samples as the positive samples, and 32 out-class samples to create adversarial examples which will be used as negative samples.

All binary classifier models use a neural network consisting of two convolutional layers each with 32 and 64 filters, and a fully connected layer of size 1024; more details of the network architecture can be found in Madry et al. (2017).

Table 5: Training setups for MNIST detection models

| | $L_2$ models | | $L_\infty$ models | |
|---|---|---|---|---|
| | $\epsilon = 2.5$ | $\epsilon = 5.0$ | $\epsilon = 0.3$ | $\epsilon = 0.5$ |
| PGD attack steps, step-size (training) | 100, 0.1 | 200, 0.1 | 100, 0.01 | 100, 0.01 |
| PGD attack steps, step-size (validation) | 200, 0.1 | 200, 0.1 | 200, 0.01 | 200, 0.01 |

## A.2 CIFAR10 TRAINING

We train CIFAR10 binary classifiers using a ResNet model (same as the one used by Madry et al. (2017); MadryLab (a)). To speedup training, we take advantage of a clean trained classifier: the subnetwork of $f$ that defines the output logit $z_f(\cdot)_k$ is essentially a "binary classifier" that would output high values for samples of class $k$, and low values for others. The binary classifier is then trained by finetuning the subnetwork using objective 5. The pretrained classifier has a test accuracy of 95.01% (fetched from MadryLab (a)).

At each iteration of training we sample a batch of 300 samples, from which in-class samples are used as positive samples, while an equal number of out-of-class samples are used for crafting adversarial examples. Adversarial examples for training $L_2$ and $L_\infty$ models are both optimized using normalized steepest descent based PGD attacks (MadryLab, b). We report results based on the best performances on the CIFAR10 test set (thus don't claim generalization performance of the proposed method).

## B COMPUTING MEAN $L_2$ DISTANCE

We first find the detection threshold $T$ with which the detection system has 0.95 TPR. We construct a new loss function by adding a weighted loss term that measures perturbation size to objective 14

$$L(x') = -\max_{i \neq y} z_H(x')_i + c \cdot \|x' - x\|_2^2. \tag{21}$$

We then use *unconstrained* PGD attack to optimize $L(x')$. We use binary search to find the optimal $c$, where in each bsearch attempt if $x'$ is a false positive ($\max_i z_H(x')_i \neq y$ and $\max_{i \neq y} z_H(x')_i > T$) we consider the current $c$ as effective and continue with a larger $c$. The configurations for performing binary search and PGD attack are detailed in Table 6. The $c$ upper bound is established such that with this upper bound, no samples except those that are inherently misclassified by the generative classifier, could be perturbed as a false positive. With these settings, our MNIST $L_\infty$ $\epsilon = 0.3$ and $L_\infty$ $\epsilon = 0.5$ generative detection models respective reached 1.0 FPR and 0.9455 FPR, and CIFAR10 generative model reached 0.9995 FPR.

Table 6: Binary search and PGD attack configurations on MNIST and CIFAR10 dataset

| Dataset | Initial $c$ | $c$ lower bound | $c$ upper bound | bsearch depth | PGD steps | PGD step size | Threshold | PGD optimizer |
|---|---|---|---|---|---|---|---|---|
| MNIST | 0.0 | 0.0 | 8.0 | 20 | 1000 | 1.0 (0-1 scale) | 3.6 | Adam |
| CIFAR10 | 0.0 | 0.0 | 1.0 | 20 | 100 | 2.56 (0-255 scale) | -5.0 | $L_2$ normalized steepest descent |

## C MORE EXPERIMENTAL RESULTS

### C.1 MORE MNIST RESULTS

Table 7: AUC scores of the first two binary classifiers ($d_1, d_2$, MNIST) tested with different configurations of PGD attacks. In each step of the PGD attack we use normalized gradient as in Madry et al. (2017) (the update rules for $L_2$-based and $L_\infty$-based attacks are respectively $x_{n+1} = x_n - \gamma \frac{\nabla f(x_n)}{\|\nabla f(x_n)\|_2}$ and $x_{n+1} = x_n - \gamma \cdot \text{sign}(\nabla f(x_n))$).

| PGD attack steps, step size | $L_\infty$ $\epsilon = 0.3$ model | | $L_\infty$ $\epsilon = 0.5$ model | | PGD attack steps, step size | $L_2$ $\epsilon = 2.5$ model | | $L_2$ $\epsilon = 5.0$ model | |
|---|---|---|---|---|---|---|---|---|---|
| | $d_1$ | $d_2$ | $d_1$ | $d_2$ | | $d_1$ | $d_2$ | $d_1$ | $d_2$ |
| 200, 0.01 | 0.99962 | 0.99973 | 0.99820 | 0.99901 | 200, 0.1 | 0.99906 | 0.99916 | 0.99960 | 0.99997 |
| 2000, 0.005 | 0.99959 | 0.99971 | 0.99795 | 0.99872 | 2000, 0.05 | 0.99855 | 0.99883 | 0.99237 | 0.99994 |

Table 8: AUC scores of the first two binary classifiers ($d_1, d_2$, MNIST) under cross-norm and cross-perturbation attacks. $L_\infty$-based attacks use steps 200 and step-size 0.01, and $L_2$-based attacks uses steps 200 and step-size 0.1.

| | $d_1$ | | | | $d_2$ | | | |
|---|---|---|---|---|---|---|---|---|
| Attack | $L_\infty$ $\epsilon = 0.3$ | $L_\infty$ $\epsilon = 0.5$ | $L_2$ $\epsilon = 2.5$ | $L_2$ $\epsilon = 5.0$ | $L_\infty$ $\epsilon = 0.3$ | $L_\infty$ $\epsilon = 0.5$ | $L_2$ $\epsilon = 2.5$ | $L_2$ $\epsilon = 5.0$ |
| $L_\infty$ $\epsilon = 0.3$ | 0.99959 | 0.99966 | 0.99927 | 0.99925 | 0.99971 | 0.99967 | 0.99949 | 0.99984 |
| $L_\infty$ $\epsilon = 0.5$ | 0.99436 | 0.9983 | 0.99339 | 0.99767 | 0.99778 | 0.99869 | 0.99397 | 0.99961 |
| $L_2$ $\epsilon = 2.5$ | 0.99974 | 0.99969 | 0.99962 | 0.99944 | 0.99965 | 0.99955 | 0.99968 | 0.99987 |
| $L_2$ $\epsilon = 5.0$ | 0.96421 | 0.98816 | 0.97747 | 0.99577 | 0.98268 | 0.98687 | 0.98117 | 0.99986 |

Table 9: AUC scores of $d_1$ (MNIST) under fixed start and multiple random restarts attacks. The $L_\infty$ $\epsilon = 0.5$ model is attacked using $L_\infty$ $\epsilon = 0.5$ constrained PGD attack of steps 200 and step size 0.01, and the $L_2$ $\epsilon = 5.0$ model is attacked using $L_2$ $\epsilon = 5.0$ constrained PGD attack of steps 200 and step size 0.1.

| | $L_\infty$ $\epsilon = 0.5$ model | $L_2$ $\epsilon = 5.0$ model |
|---|---|---|
| fixed start | 0.99830 | 0.99578 |
| 50 random restarts | 0.99776 | 0.99501 |

Table 10: AUC scores of all $L_\infty$ $\epsilon = 0.3$ binary classifiers. Results obtained with $L_\infty$ $\epsilon = 0.3$ constrained PGD attacks of steps 200 and step size 0.01.

| | $d_1$ | $d_2$ | $d_3$ | $d_4$ | $d_5$ | $d_6$ | $d_7$ | $d_8$ | $d_9$ | $d_{10}$ |
|---|---|---|---|---|---|---|---|---|---|---|
| AUC | 0.99959 | 0.99971 | 0.99876 | 0.99861 | 0.99859 | 0.99861 | 0.99795 | 0.99863 | 0.99687 | 0.99418 |

Table 11: AUC scores of all $L_\infty$ $\epsilon = 0.5$ binary classifiers. Results obtained with $L_\infty$ $\epsilon = 0.5$ constrained PGD attacks of steps 200 and step size 0.01.

| | $d_1$ | $d_2$ | $d_3$ | $d_4$ | $d_5$ | $d_6$ | $d_7$ | $d_8$ | $d_9$ | $d_{10}$ |
|---|---|---|---|---|---|---|---|---|---|---|
| AUC | 0.99830 | 0.99869 | 0.99327 | 0.99355 | 0.99314 | 0.99228 | 0.99424 | 0.99439 | 0.97875 | 0.9769 |

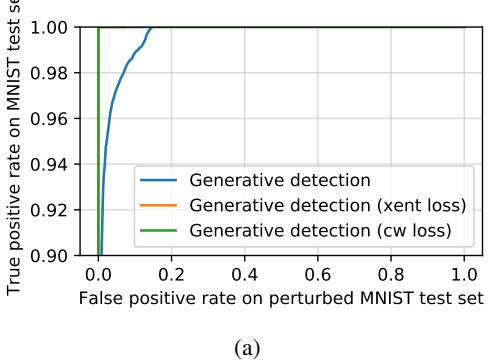

(a)

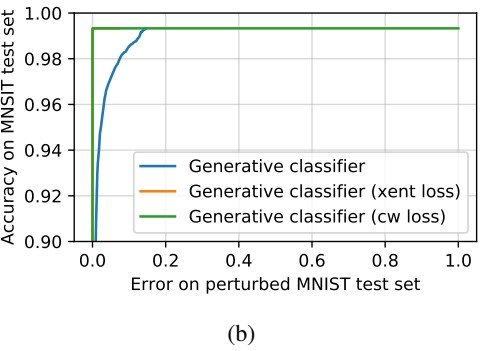

(b)

Figure 8: Performance of generative detection (a) and generative classification (b) on MNIST dataset under attacks with different loss functions. Please refer to MadryLab (b) for the implementations of cross-entropy loss and CW loss based attacks.

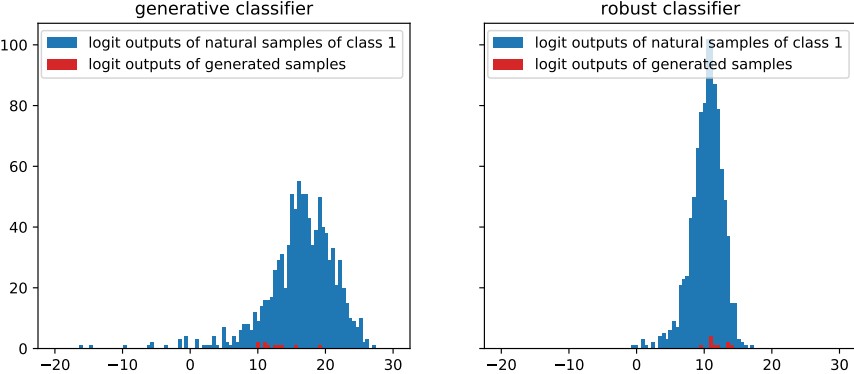

Figure 9: Distributions of class 1's logit outputs of clean samples from class 1 and perturbed samples from the first row of Figure 4 (MNIST dataset).

## C.2 MORE CIFAR10 RESULTS

### C.2.1 MORE ROBUSTNESS TEST RESULTS

Table 12: AUC scores of CIFAR10 $d_1$ under fixed start and multiple random restarts attacks. The $L_\infty$ $\epsilon = 2.0$ model is attacked using PGD attack of steps 10 and step size 0.5, and the $L_\infty$ $\epsilon = 8.0$ model is attacked using PGD attack of steps 40 and step size 0.5.

|  | $L_\infty$ $\epsilon = 2.0$ model | $L_\infty$ $\epsilon = 8.0$ model |
|---|---|---|
| fixed start | 0.9866 | 0.9234 |
| 10 random starts | 0.9866 | 0.9233 |

Table 13: AUC scores of CIFAR10 $d_1$ (trained with $L_\infty$ $\epsilon = 8$) under PGD attacks with different norms and perturbation limits.

| PGD attack | AUC |
|---|---|
| $L_2$ $\epsilon = 80$-constrained (steps 20, step-size 10) | 0.9814 |
| $L_\infty$ $\epsilon = 2$-constrained (steps 10, step-size 0.5) | 0.9841 |

Table 14: AUC scores of CIFAR10 $d_1, d_2$ (trained with $L_2$ $\epsilon = 80$-constrained PGD attack of steps 20 and step size 10) under $L_2$-based PGD attacks of different steps and step-size.

| $L_2$ $\epsilon = 80$-constrained PGD attack steps, step size | $d_1$ | $d_2$ |
|---|---|---|
| 20, 10 | 0.9839 | 0.9924 |
| 50, 5.0 | 0.9837 | 0.9922 |

Table 15: AUC scores of all CIFAR10 $L_\infty$ $\epsilon = 2.0$ binary classifiers under $L_\infty$ $\epsilon = 2.0$-constrained PGD attack of steps 10 and step size 0.5.

|  | $d_1$ | $d_2$ | $d_3$ | $d_4$ | $d_5$ | $d_6$ | $d_7$ | $d_8$ | $d_9$ | $d_{10}$ |
|---|---|---|---|---|---|---|---|---|---|---|
| AUC | 0.9866 | 0.9926 | 0.9721 | 0.9501 | 0.9773 | 0.9636 | 0.9859 | 0.9908 | 0.9930 | 0.9916 |

Table 16: AUC scores of all CIFAR10 $L_\infty$ $\epsilon = 8.0$ binary classifiers under $L_\infty$ $\epsilon = 8.0$-constrained PGD attack of steps 40 and step size 0.5.

|  | $d_1$ | $d_2$ | $d_3$ | $d_4$ | $d_5$ | $d_6$ | $d_7$ | $d_8$ | $d_9$ | $d_{10}$ |
|---|---|---|---|---|---|---|---|---|---|---|
| AUC | 0.9234 | 0.9553 | 0.8393 | 0.7893 | 0.8494 | 0.8557 | 0.9071 | 0.9276 | 0.9548 | 0.9370 |

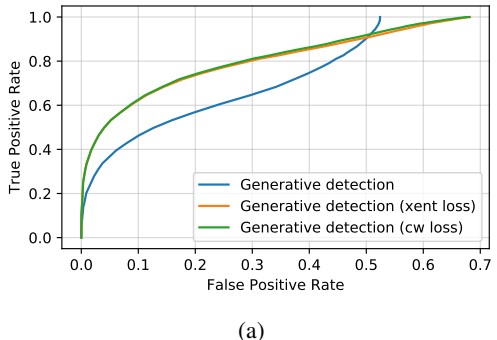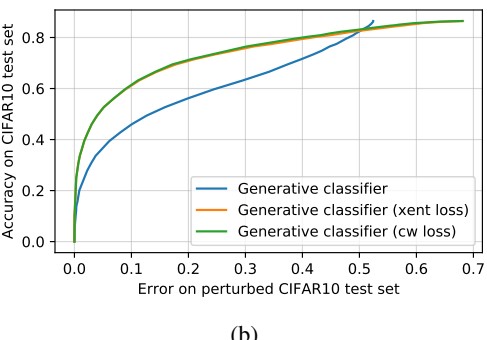

(a)                                                    (b)

Figure 10: Performance of generative detection (a) and generative classification (b) on CIFAR10 dataset under attacks with different loss functions. Cross-entropy and CW loss is only able to outperforms loss 14 when detection threshold is low (over 0.9 TPR). Please refer to MadryLab (a) for the implementations of cross-entropy loss and CW loss based attacks.

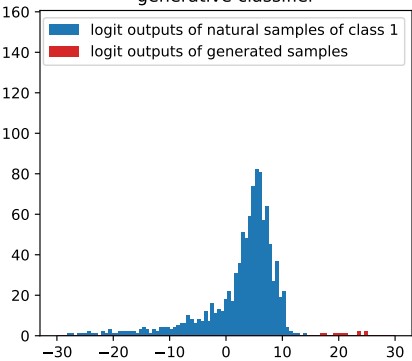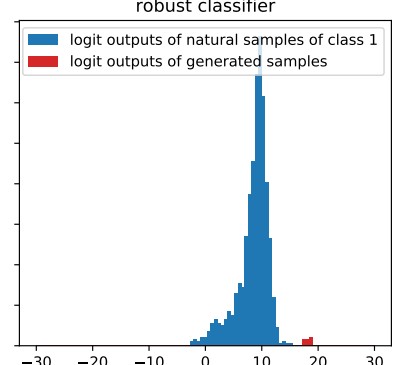

Figure 11: Distributions of class 1's logit outputs of clean samples of class 1 and generated samples from the first row of Figure 7 (CIFAR10 dataset).

### C.2.2 TRAINING STEP SIZE AND ROBUSTNESS

We found training with adversarial examples optimized with a sufficiently small step size to be critical for model robustness. In table 17 we tested two $L_\infty$ $\epsilon = 2.0$ binary classifiers respectively trained with 0.5 and 1.0 step size. The step size 1.0 model is not robust when tested with a much smaller step size. We observe that when training the step size 1.0 model, training set adv AUC reached 1.0 in less than one hundred iterations, but test set clean AUC plummeted to around 0.95 and couldn't recover thereafter. (Please refer to Figure 12 for the definitions of adv AUC and nat AUC.)

Table 17: AUC scores of two $L_\infty$ $\epsilon = 2.0$ $d_1$ models trained with different steps and step-sizes.

| Attack steps, step-size | Training steps, step-size | |
| --- | --- | --- |
| | 10, 0.5 | 10, 1.0 |
| 10, 0.5 | 0.9866 | 0.9965 |
| 40, 0.1 | 0.9892 | 0.8848 |

### C.2.3 EFFECTS OF PERTURBATION LIMIT

To study the effects of perturbation limit, we analyze the training dynamics of one $L_\infty$ $\epsilon = 2.0$ constrained and one $L_\infty$ $\epsilon = 8.0$ constrained binary classifiers. In Figure 12 we show the training and testing history of these two models. The $\epsilon = 2.0$ model history shows that by adversarial finetuning the model reaches robustness in just a few thousands of iterations, and the performance on clean samples is preserved (test clean AUC begins at 0.9971, and ends at 0.9981). Adversarial finetuning on the $\epsilon = 8.0$ model didn't converge after an extended 20K iterations of training. The gap between train adv AUC and test adv AUC of the $\epsilon = 8.0$ model is more pronounced, and we observed a decrease of test clean AUC from 0.9971 to 0.9909.

These results suggest that training with larger perturbation limit is more time and resource consuming, and could lead to performance decrease on clean samples. The benefit is that the detector is pushed to a better approximation of the target data distribution. As an illustration, in Figure 13, perturbations generated by attacking the naturally trained classifier (corresponds to 0 perturbation limit) don't have clear semantics, while perturbed samples of the $L_\infty$ $\epsilon = 8.0$ model are completely recognizable.

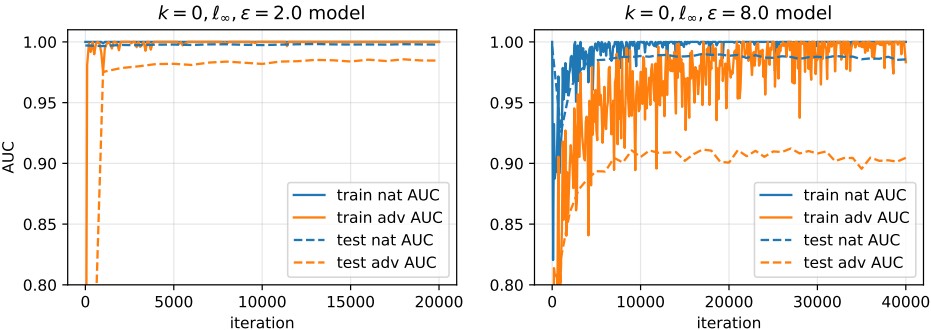

Figure 12: Training and testing AUC histories of two binary classifiers. Adv AUC is the AUC score computed on $\{(x, 0) : x \in \mathcal{D}'^f_{\setminus k}\} \cup \{(x, 1) : x \in \mathcal{D}^f_k\}$, and nat AUC is the score computed on $\{(x, 0) : x \in \mathcal{D}^f_{\setminus k}\} \cup \{(x, 1) : x \in \mathcal{D}^f_k\}$.

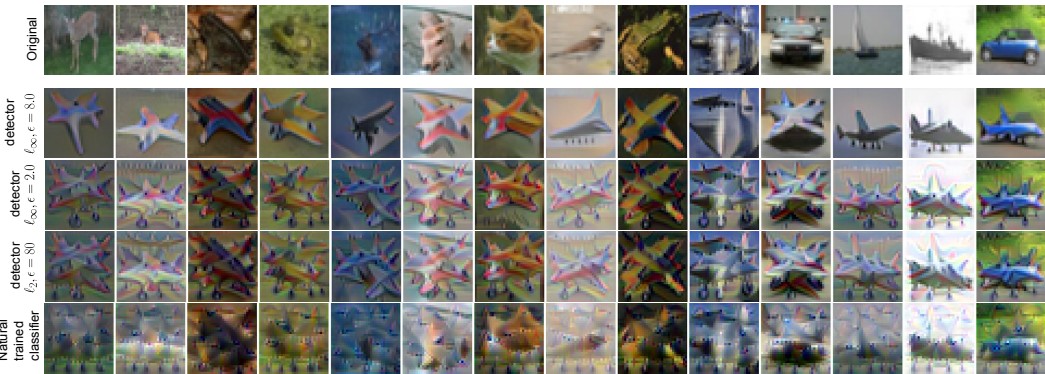

Figure 13: Perturbed samples produced by attacking the $k = 0$ (airplane) detectors and the natural trained classifier's 1st logit output. All samples reached the same $L_2$ perturbation of 1200 (produced using PGD attacks of step size 10.0).

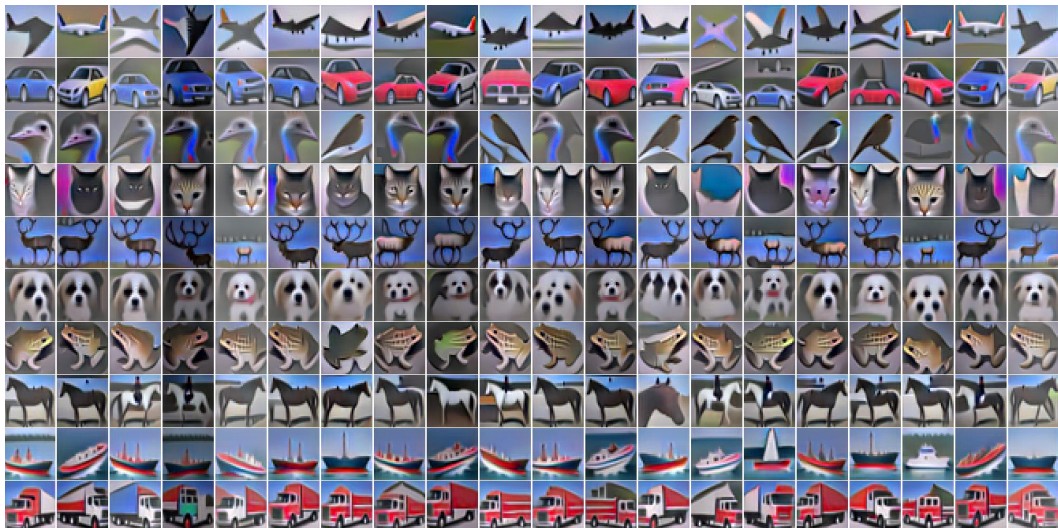

Figure 14: Images generated from class conditional Gaussian noise by attacking $L_\infty$ $\epsilon = 16$ constrained CIFAR10 detectors. we use $L_2$ $\epsilon = 100 \times 255$ constrained PGD attack of steps 200 and step size $0.5 \times 255$. Samples not selected.

## C.3 IMAGENET RESULTS

On ImageNet we show GAT induces detection robustness and supports the learning of class conditional distributions. Our experiment is based on Restricted ImageNet (Tsipras et al., 2018), a subset of ImageNet that has its samples reorganized into customized categories. The dog category consists of images of different breeds collected from ImageNet class from 151 to 268. We trained a dog class detector by finetuning a pre-trained ResNet50 (He et al., 2016) model. The dog category covers a range of ImageNet classes, with each one having its logit output. We use the subnetwork defined by the logit output of class 151 as the detector (in principle logit output of other classes in the range should also work). Due to computational resource constraints, we only validated the robustness of a $L_\infty$ $\epsilon = 0.02$ trained detector (trained with PGD attack of steps 40 and step size 0.001), and we present the result in Table 18. (On Restricted ImageNet in the case of $L_\infty$ scenario Tsipras et al. (2018) only demonstrates the robustness of a $\epsilon = 0.005$ constrained model). Please refer to Appendix C.3 for more results on adversarial example generation and image synthesis.

Table 18: AUC scores of the dog detector under different strengths of $L_\infty$ $\epsilon = 0.02$ constrained PGD attacks

| Attack steps, step size | 40, 0.001 | 100, 0.001 | 200, 0.001 | 40, 0.002 | 200, 0.002 | 200, 0.0005 |
|---|---|---|---|---|---|---|
| AUC | 0.9720 | 0.9698 | 0.9692 | 0.9703 | 0.9690 | 0.9698 |

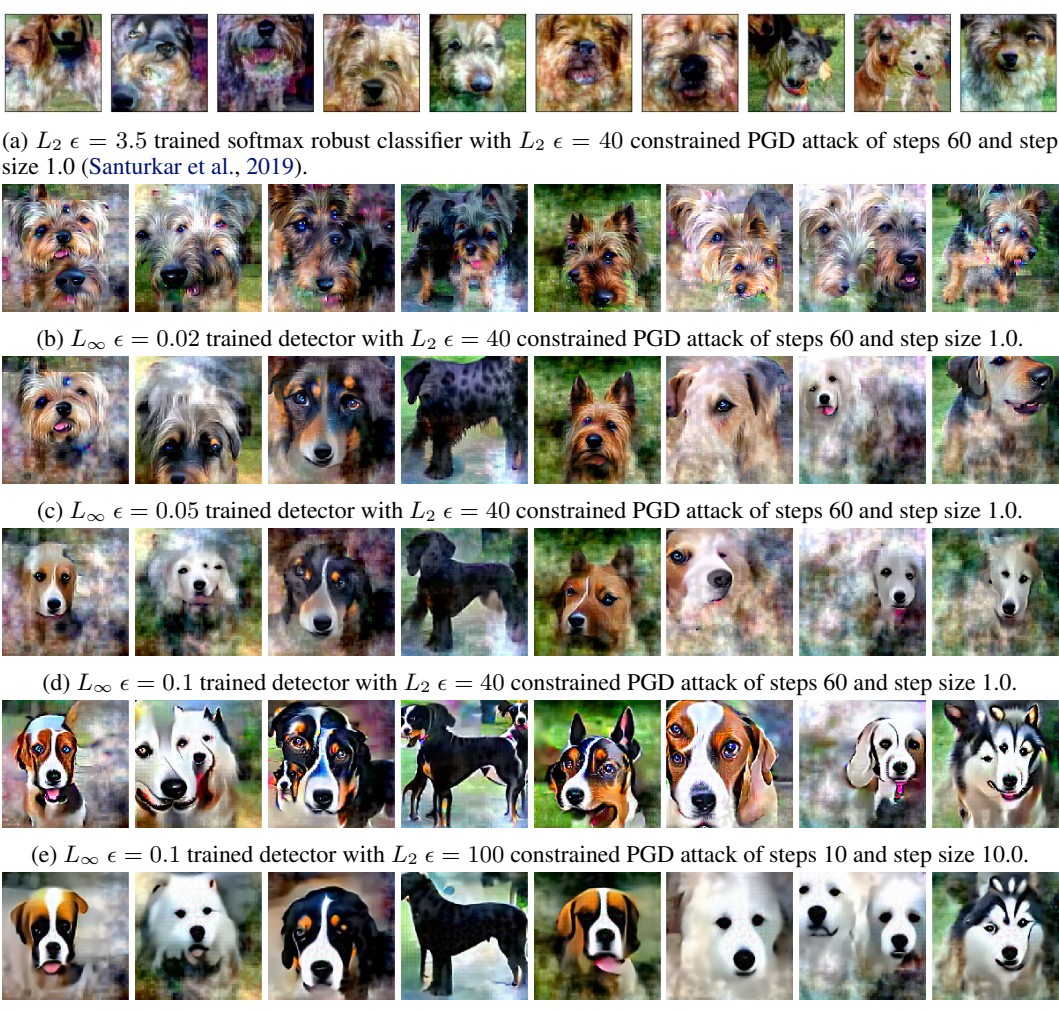

(a) $L_2$ $\epsilon = 3.5$ trained softmax robust classifier with $L_2$ $\epsilon = 40$ constrained PGD attack of steps 60 and step size 1.0 (Santurkar et al., 2019).

(b) $L_\infty$ $\epsilon = 0.02$ trained detector with $L_2$ $\epsilon = 40$ constrained PGD attack of steps 60 and step size 1.0.

(c) $L_\infty$ $\epsilon = 0.05$ trained detector with $L_2$ $\epsilon = 40$ constrained PGD attack of steps 60 and step size 1.0.

(d) $L_\infty$ $\epsilon = 0.1$ trained detector with $L_2$ $\epsilon = 40$ constrained PGD attack of steps 60 and step size 1.0.

(e) $L_\infty$ $\epsilon = 0.1$ trained detector with $L_2$ $\epsilon = 100$ constrained PGD attack of steps 10 and step size 10.0.

(f) $L_\infty$ $\epsilon = 0.3$ trained detector with $L_2$ $\epsilon = 100$ constrained PGD attack of steps 100 and step size 10.0.

Figure 15: ImageNet $224 \times 224 \times 3$ random samples generated from class conditional Gaussian noise by attacking softmax robust classifier and detector models trained with different constrains. Note than large perturbation models didn't reach robustness. Please refer to Santurkar et al. (2019) for the detail about how the class conditional Gaussian is estimated.

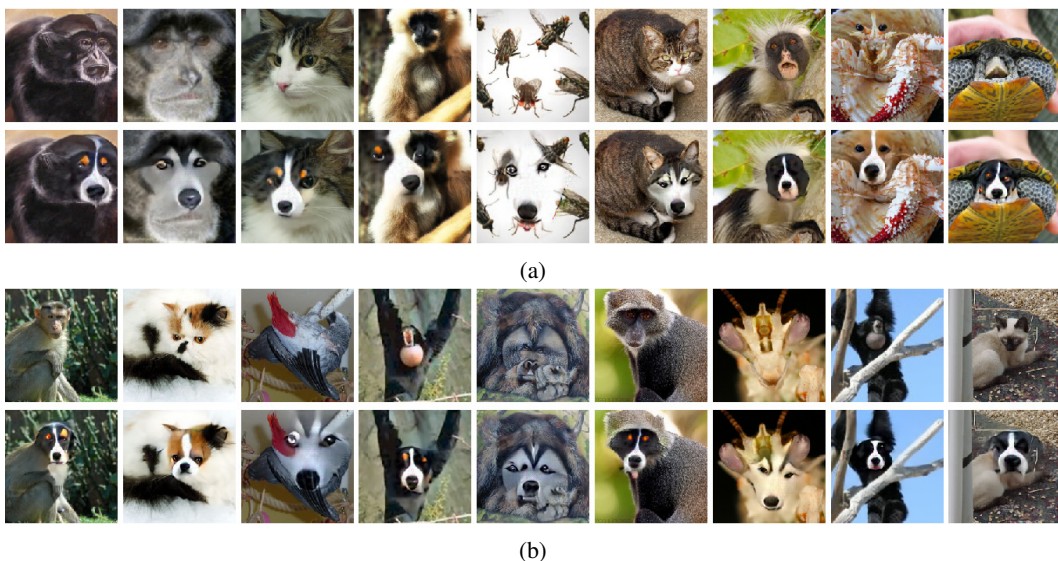

Figure 16: Perturbed samples produced by attacking the $L_\infty$ $\epsilon = 0.3$ trained dog detector using $L_2$ $\epsilon = 30$ constrained PGD attack of steps 100 and step size 5. Top rows are original images, and second rows are attacked images.

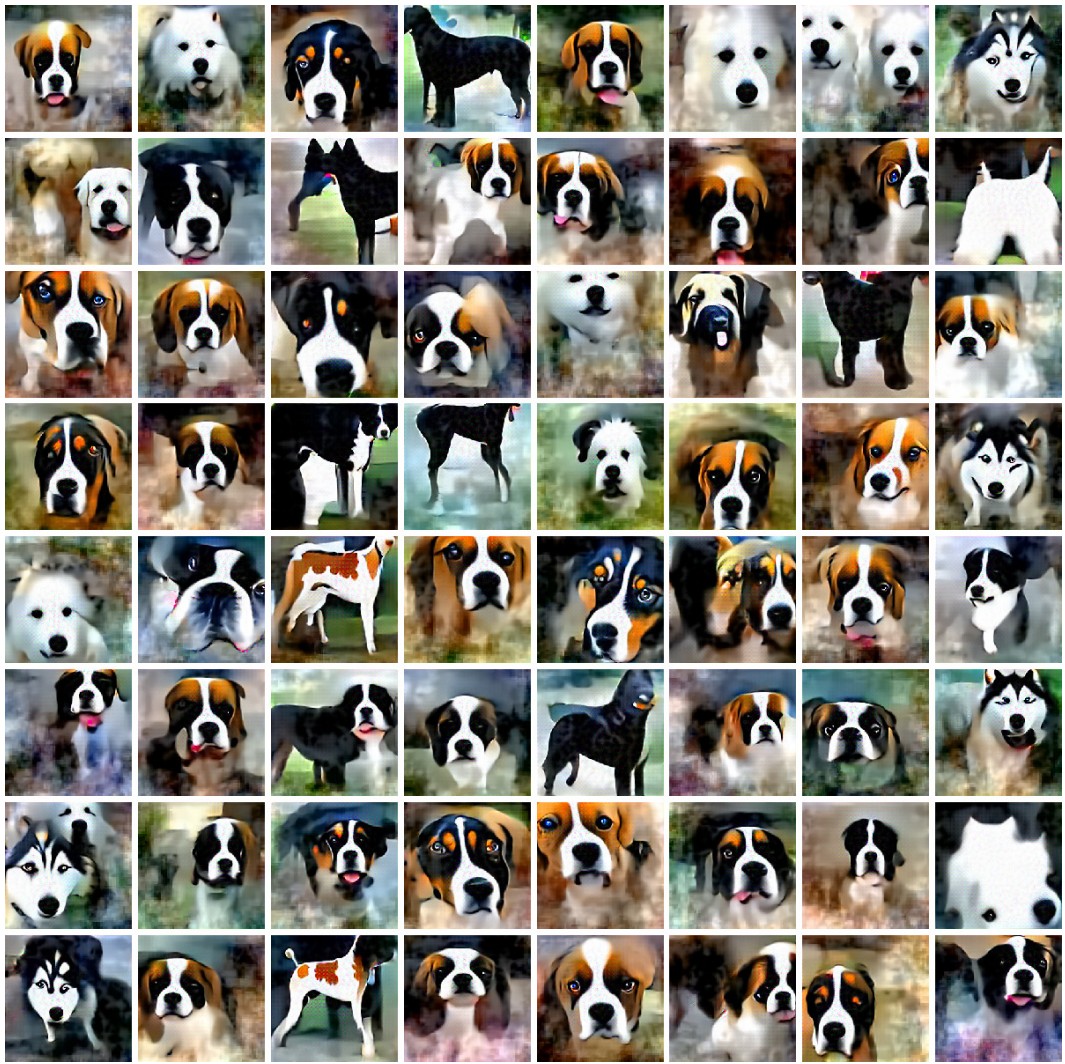

Figure 17

Figure 18: More $224 \times 224 \times 3$ random samples generated by attacking the $L_\infty$ $\epsilon = 0.3$ trained detector with $L_2$ $\epsilon = 100$ constrained PGD attack of steps 100 and step size 10.0.

## D  Gaussian noise attack and model interpretability

In this section we use Gaussian noise attack experiment to motivate a comparative analysis of the interpretabilities of our generative classification approach and discriminative robust classification approach (Madry et al., 2017).

We first discuss how these two approaches determine the posterior class probabilities.

For the discriminative classifier, the posterior probabilities are computed from the logit outputs of the classifier using the softmax function $p(k|x) = \frac{\exp(z_f(x)_k)}{\sum_{j=1}^{K} \exp(z_f(x)_j)}$. For the generative classifier, the posterior probabilities are computed in two steps: in the first, we train the base detectors, which is the process of solving the inference problem of determining the joint probability $p(x, k)$, and in the second, we use Bayes rule to compute the posterior probability $p(k|x) = \frac{p(x,k)}{p(x)} = \frac{\exp(z_{d_k}(x))}{\sum_{j=1}^{K} \exp(z_{d_j}(x))}$. Coincidentally, the formulas for computing the posterior probabilities take the same form. But in our approach, the exponential of the logit output of a detector (i.e., $\exp(z_{d_k}(x))$) has a clear probabilistic interpretation: it's the *unnormalized joint probability* of the input and the corresponding class category. We use Gaussian noise attack to demonstrate that this probabilistic interpretation is consistent with visual perception.

We start from a Gaussian noise image, and gradually perturb it to cause higher and higher logit outputs. This is implemented by targeted PGD attack against logit outputs of these two classification models. The resulting images in Figure 19 show that, in our model, the logit output increase direction, i.e. the join probability increase direction, indicates the class semantic changing direction; while for the discriminative robust model, the perturbed image computed by increasing logit outputs are not as clearly interpretable. In particular, the perturbed images that cause high logit outputs of the softmax robust classifiers are not recognizable.

In summary, as a generative classification approach that explicitly models class conditional distributions, our system offers a probabilistic view of the decision making process of the classification problem; adversarial attacks that rely on imperceptible or uninterpretable noises are not effective against such a system.

## E  Computational cost issue

In this section we provide an analysis of the computational cost of our generative classification approach. In terms of memory requirements, if we assume the softmax classifier (i.e., the discriminative softmax robust classifier) and the detectors use the same architecture (i.e., only defer in the final layer) then the detector based generative classifier is approximately $K$ times more expensive than the $K$-class softmax classifier. This also means that the computational graph of the generative classifier is $K$ times larger than the softmax classifier. Indeed, in the CIFAR10 task, on our Quadro M6000 24GB GPU (TensorFlow 1.13.1), the inference speed of the generative classifier is roughly ten times slower than the softmax classifier.

We next benchmark the training speed of these two types of classifiers.

The generative classifier has $K$ logit outputs, with each one defined by the logit output of a detector. Same with the softmax classifier, except that the $K$ outputs share the parameters in the convolutional base. Now consider ordinary adversarial training on the softmax classifier and generative adversarial training on the generative classifier. To train the softmax classifier, we use batches of $N$ samples. For the generative classifier, we train each detector with batches of $2 \times M$ samples ($M$ positive samples and $M$ negative samples). At each iteration, we need to respectively compute $N$ and $M \times K$ adversarial examples for these two classifiers. Now we test the speed of the following two scenarios: 1) compute the gradient w.r.t. to N samples on a single computational graph, and 2) compute the gradient w.r.t to $M \times K$ samples on $K$ computational graphs, with each graph working on $M$ samples. We assume that in scenario 2 all the computational graphs are loaded to GPUs, and thus their computations are in parallel.

In our CIFAR10 experiment, we used batches consisting of 30 positive samples and 30 negative samples to train each ResNet50 binary classifiers. In Madry et al. (2017), the softmax classifier was trained with batches of 128 samples. In this case, $K = 10$, $M = 30$, and $N = 128$. On our GPU,

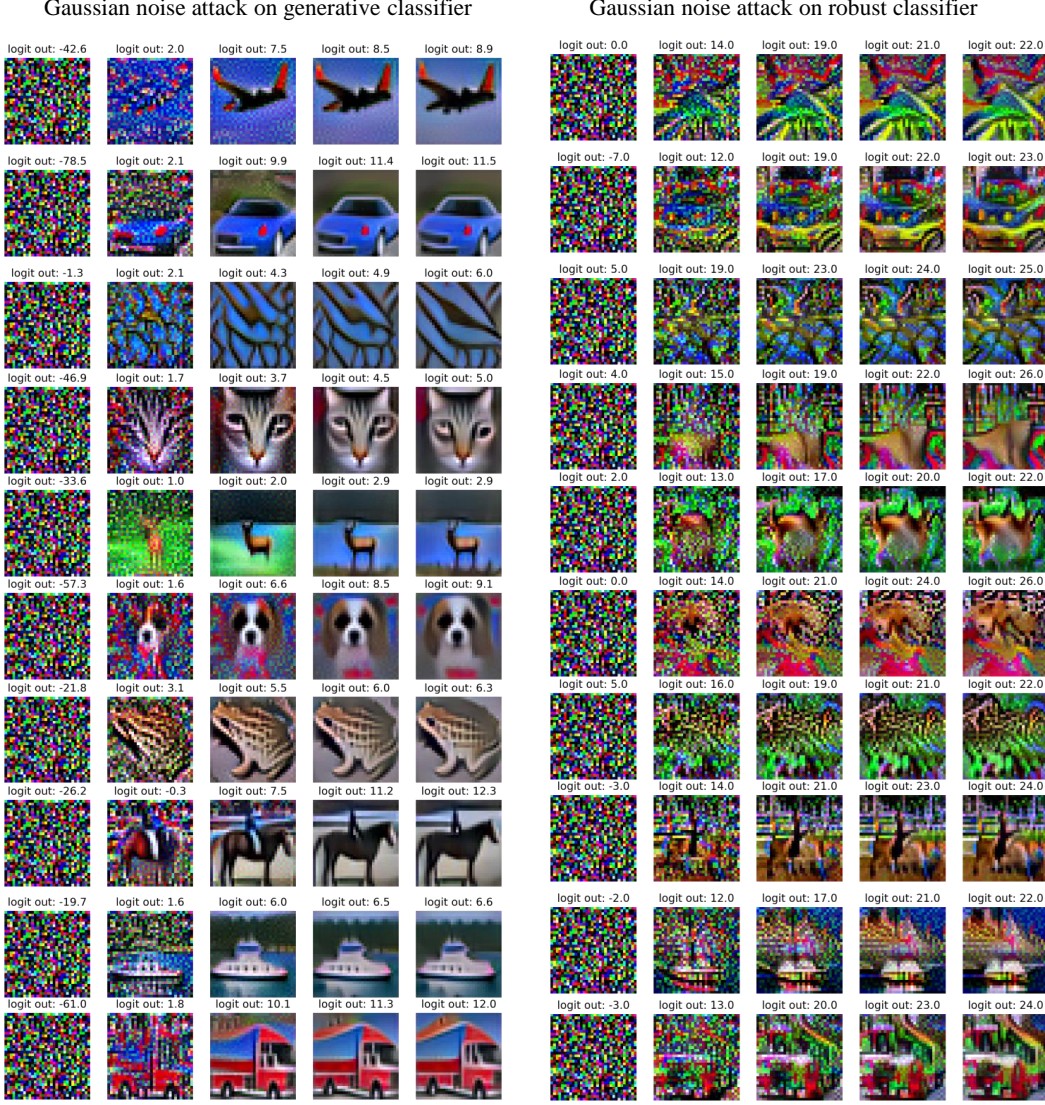

Figure 19: Image generated by attacking the generative classifier (based on $L_\infty$ $\epsilon = 16$ trained detectors) and discriminative softmax robust classifier (Madry et al., 2017) using (the same) Gaussian noise image. We used unconstrained $L_2$ PGD attack of step size 0.5*255. The five columns corresponding to the perturbed images at step 0, 50, 100, 150, and 200.

scenario 1 took 683 ms $\pm$ 6.76 ms per loop, while scenario 2 took 1.85 s $\pm$ 42.7 ms per loop. In this case, we could expect generative adversarial training to be about 2.7 times slower than ordinary adversarial training, if not considering parameter gradient computation.

In practice, large batch size is almost always preferred. And our method won't compare as favorably if we choose to use one.

## F  DENSITY ESTIMATION ON SYNTHETIC DATASETS

While ordinary discriminative training only learns a good decision boundary, GAT is able to learn the underlying density functions that generate the training data. Results on 1D (Figure 20) and 2D benchmark datasets (Figure 21) show that through properly configured generative adversarial training, detectors' output recover target density functions.

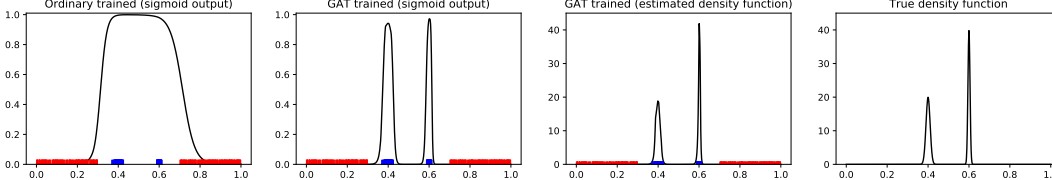

Figure 20: Ordinary discriminative training and generative adversarial training on real 1D data. The positive class data (blue points) are sampled from a mixture of Gaussians (mean 0.4 with std 0.01, and mean 0.6 with std 0.005, each with 250 samples). Both the blue and red data has 500 samples. The estimated density function is computed using Gibbs distribution and network logit outputs. PGD attack steps 20, step size 0.05, and perturbation limit $\epsilon = 0.3$.

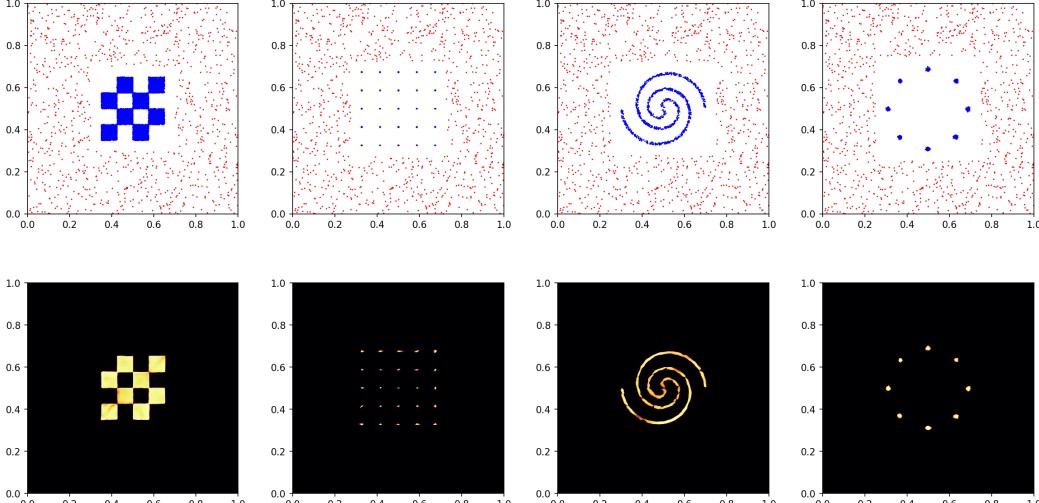

Figure 21: 2D datasets (top row, blue points are class 1 data, and red points are class 0 data, both have 1000 data points) and sigmoid outputs of GAT trained models (bottom row). The architecture of the MLP model for solving these tasks is 2-500-500-500-500-500-1. PGD attack steps 10, step size 0.05, and perturbation limit $L_\infty$ $\epsilon = 0.5$.

