# OpenReview forum: "GAT: Generative Adversarial Training for Adversarial Example Detection and Robust Classification"
_ICLR.cc/2020/Conference — Accept (Poster)_

### Official Review · AnonReviewer1 · 2019-10-20
**Official Blind Review #1**

**Rating:** 6

**Review:**

Summary:
This paper studies the adversarial detection problem within the robust optimization framework. They propose an adversarial detection and a generative modeling technique called asymmetrical adversarial training (AAT). With one detector for each class discriminating natural data from adversarially perturbed data, AAT can learn class-conditional distributions, which further result in generative detection/classification methods with competitive performance. Experimental results are provided on MNIST, CIFAR10 and Restricted ImageNet, compared with CW method as baseline.

The paper is well written with detailed experimental results. I'd suggest accepting the paper.

To my understanding, the objective function of AAT is similar to GAN's, while there is a detector for each class discriminating natural data from adversarially perturbed data instead of generated data. They incorporate the attack into the training objective with three attacking scenarios: classifier attack, detectors attack, and combined attack. They also introduce integrated classification of the classifier and detectors with the reject option. Further, they demonstrate ATT promotes the learning of class-conditional distributions and leads to generative classifiers. They claim in addition to more robust classification, ATT also gives rise to improved interpretability, which I'm not convinced of with given experimental results.

**Experience Assessment:**

I have read many papers in this area.

**Review Assessment: Checking Correctness Of Derivations And Theory:**

I assessed the sensibility of the derivations and theory.

**Review Assessment: Checking Correctness Of Experiments:**

I assessed the sensibility of the experiments.

**Review Assessment: Thoroughness In Paper Reading:**

I read the paper at least twice and used my best judgement in assessing the paper.

---

> ### Author Response · Authors · 2019-11-10
> **Improved interpretability**
>
> We thank the reviewer for a positive evaluation of our contributions and for raising a thought-provoking question. We agree that we didn't provide a proper discussion about the interpretability of our method. Here are our thoughts and some additional experimental results.
>
> We start with a discussion about how these two approaches - our generative classification approach, and the discriminative robust classification approach - determine the posterior probabilities of inputs.
>
> In our approach, the posterior is computed in two steps. In the first step, we train our base detectors, which is the process of solving the inference problem of determining the joint probability $p(x,k)$. Unfortunately, the detectors don't define explicit density functions. We could, however, use the logit output of the detector to define an energy function [1] $E_{\theta_k}(x) = -z(h_k(x))$, and then use the Gibbs distribution to obtain the joint probability $p(x, k) = \frac{\exp(-E_{\theta_k}(x))}{Z_\Theta}=\frac{\exp(z(h_k(x)))}{Z_\Theta}$, where  $Z_\Theta$ is a normalizing constant known as the partition function, and could be computed in our case as $Z_\Theta = \sum_k\int\exp(-E_{\theta_k}(x))dx$. (We note that a concurrent ICLR2020 submission https://openreview.net/forum?id=Hkxzx0NtDB uses a similar formulation.) In the second step we use Bayes rule to compute the posterior probability $p(k|x) = p(x,k)/p(x) = \frac{\exp(z(h_k(x)))}{\sum_{j=1}^K \exp(z(h_j(x)))}$. Although here $Z_\Theta$ is intractable, it doesn't appear in the posterior, so knowing the logit outputs of the base detectors will suffice. (The readers may have noticed that the above formulation is different from the one presented in our manuscript. It's because after we later realized that we are not able to do explicit density estimation, we resorted to the energy-based learning framework [1], and worked out an energy-based solution. Note that our experimental results are not affected by the reformulation. In our updated manuscript we provide more data and discussions to justify the energy-based formulation.)
>
> In the discriminative classification approach, the posterior probability is computed from the logit outputs of the classifier using the softmax function $p(k|x) = \frac{\exp(z(f(x))_k)}{\sum_{j=1}^K \exp(z(f(x))_j)}$.
>
> Coincidentally, the formulas for computing the posterior probabilities take the same form. But in our approach, the exponential of the logit output of a detector (i.e., $\exp(z(h_k(x)))$) has a clear probabilistic interpretation: it's the *unnormalized joint probability* of the input and the corresponding class category. We now demonstrate that this probabilistic interpretation is consistent with our visual perception.
>
> We start from a Gaussian noise image, and gradually perturb it to cause higher and higher logit outputs. This is implemented by targeted PGD attack against logit outputs of corresponding models. The resulting images are in the last figure of this README file https://github.com/nhLKeO/AAT-CIFAR10/blob/master/README.md . It's clear that for our model, the logit output increase direction is the semantic changing direction; while for the discriminative robust model, the perturbed image computed by increasing logit outputs are not as clearly interpretable. In particular, the perturbed images that cause high logit outputs of the robust classifiers are not recognizable.
>
> We hope that our explanation could convince you that our approach improves the interpretability by providing a probabilistic view of the decision process of the classification problem.
>
> [1] LeCun, Yann, et al. "A tutorial on energy-based learning." *Predicting structured data* 1.0 (2006).

---

### Official Review · AnonReviewer2 · 2019-10-23
**Official Blind Review #2**

**Rating:** 6

**Review:**

Review: The paper addresses the adversarial example detection problem. The framework proposed in the paper divides the input space into subspaces based on a classifier’s output and trains detectors on these subspaces to classify a natural sample (classified as that class) from an adversarial one fooling the network. The goal is to use a robust optimization approach to enable detection methods to withstand adaptive/dynamic attacks. Hence, an asymmetrical adversarial training (AAT) regime is employed which presents solving a min-max problem. AAT supports the detectors to learn class conditional distributions, motivating generative detection/classification approaches. There are three different attacking scenarios and evaluation shows that the combined attack turns out to be most effective (as it fools both the classifier and detectors) against integrated detection. The paper also demonstrates empirical improvements over state of the art detection techniques with higher L2 distortion of perturbed samples.

Pros:
- With the vulnerabilities associated with neural networks, the motivation behind building defense mechanisms against adversarial attacks has been well-justified.
- Most of the evaluation metrics look appropriate and well-defined.
- It was interesting to observe the perturbed samples produced by attacking generative classifier. While they exhibited visible features of the target class, these perturbations had to be different on a sematic level to be distinguished from the natural samples.


Cons:
- While the idea to partition into subspaces and learn a different detection for each of them is novel, it involves training multiple detectors one by one that can be computationally expensive. The loss function for different attacking scenarios uses the outputs from all the detectors, which can also be expensive, especially when there are a lot of classes.
- To deal with extremely unbalanced data sets when training the detector, the solutions used in the paper resamples to balanced the positive and negative class data. This would mean throwing off most of the data, I would see how it affects the training.


**Experience Assessment:**

I do not know much about this area.

**Review Assessment: Checking Correctness Of Derivations And Theory:**

I assessed the sensibility of the derivations and theory.

**Review Assessment: Checking Correctness Of Experiments:**

I carefully checked the experiments.

**Review Assessment: Thoroughness In Paper Reading:**

I read the paper thoroughly.

---

> ### Author Response · Authors · 2019-11-10
> **Author Response to Official Blind Review #2**
>
> Thank you very much for your insightful review! We provide our answers below.
>
> >While the idea to partition into subspaces and learn a different detection for each of them is novel, it involves
> >training multiple detectors one by one that can be computationally expensive. The loss function for different
> >attacking scenarios uses the outputs from all the detectors, which can also be expensive, especially when there
> >are a lot of classes.
>
> We agree with the reviewer that having a detector per class could could incur high computational cost, especially for problems with a large number of classes. Your thoughtful question has motivated us to think deeper about the computational issue and do additional benchmarking. Below we summarize our points.
>
> In terms of memory requirements, if we assume the softmax classifier and the detectors use the same architecture (i.e., only defer in the final layer) then the detector-based generative classifier is approximately K times more expensive than the  K-class softmax classifier. This also means that the computational graph of the generative classifier is K times larger than the softmax classifier. Indeed, in the CIFAR10 task, on our Quadro M6000 24GB GPU (TensorFlow 1.13.1), the inference speed of the generative classifier is roughly ten times slower than the softmax classifier.  However, we would like to point out that for our "integrated detection" approach, inference only requires two times computation (first obtain the ordinary classifier's prediction, then run the corresponding detector).
>
> We next benchmark the training speed of these two types of classifiers.
>
> The generative classifier has K logit outputs, with each one defined by the logit output of a detector. Same with the softmax classifier, except that the K outputs share the parameters in the convolutional part. Now consider ordinary adversarial training on the softmax classifier and asymmetrical adversarial training on the generative classifier. To train the softmax classifier, we use batches of N samples. For the generative classifier, we train each detector with batches of 2*M samples (M positive samples + M negative samples). At each iteration, we need to respectively compute N and M*K adversarial examples for these two classifiers. Now we test the speed of the following two scenarios:  1) compute the gradient w.r.t. to N samples on a single computational graph, and 2) compute the gradient w.r.t to M*K samples on K computational graphs, with each graph working on M samples. We assume in scenario 2 that all the computational graphs are loaded to GPUs, and their computations are in parallel.
>
> In our CIFAR10 experiment, we used batches consisting of 30 positive samples and 30 negative samples to train each ResNet50 based detector. In Madry et al., the softmax classifier was trained with batches of 128 samples. In this case, K=10, M=30, and N=128.  On our GPU, scenario 1 took 683 ms ± 6.76 ms per loop, while scenario 2 took 1.85 s ± 42.7 ms per loop. So in this case, we could expect asymmetrical adversarial training to be about 2.7 times slower than ordinary adversarial training, if not considering parameter gradient computation. (If we choose to use a large batch size for the sake of more stable training, the computational cost will increase accordingly.)
>
> Another factor to take into consideration is that we extensively use adversarial finetuning (training starts from a naturally trained model). We are not sure why this practice is not exercised in ordinary adversarial training, but we found it to be an effective trick for speeding up training (see Appendix "Effects of perturbation limit").
>
> A straightforward solution to the computational issue would be to let detectors share their convolutional kernels. If we allow our detectors to share the same trunk and allocate different heads for the detectors, we will enhance the computational cost of the generative classifier.  The architecture of the generative classifier could also be made to be precisely the same as the softmax classifier (imagine replacing cross-entropy loss with binary cross-entropy loss).  We are currently looking into this new direction and will update our paper if we find this to be a viable solution.
>
> Notebook to produce these results: https://github.com/nhLKeO/AAT-CIFAR10/blob/master/speed_test.ipynb
>
> >To deal with extremely unbalanced data sets when training the detector, the solutions used in the paper
> >resamples to balanced the positive and negative class data. This would mean throwing off most of the data, I
> >would see how it affects the training.
>
> It's true that in each batch we drop some negative samples (more specifically, we only utilize the first n negative sample where n is the number of positive samples in the current batch), but at the same time we randomly shuffle the whole training set at the beginning of each epoch. So the overall effect is that each negative sample has equal probability of being dropped (or visited).

---

### Official Review · AnonReviewer3 · 2019-10-26
**Official Blind Review #3**

**Rating:** 6

**Review:**

This paper presents a method for adaptive adversarial example detection. The authors propose to construct the adversarial subspace detector based on Asymmetrical Adversarial Training (AAT). The proposed model is composed of both classifier and adversarial detector, where the classifier makes the classification prediction and the adversarial detector evaluate if the input sample is natural of adversarial. The goal of the objective function is to minimize the adversarial detector error given large enough perturbation budget.

The authors provide extensive experimental results showing the promising performance of the model in detecting various types of adversarial attack. I have several concerns regarding the model and experiments:

1) Since D^{'^{f}}_k \subset D^f_k, would the model minimize w.r.t. both the loss of L(h(x, \theta), 0) and L(h(x, \theta), 1)? Would this cause unstable training?

2) Maybe I missed it, but it seems that the objective function in Eq. (5) is based on the adversarial detector. How could the classification performance of classifier f be guaranteed in training?

3) What does the cross mark mean in Fig. 2(b) and 4(b)?

**Experience Assessment:**

I do not know much about this area.

**Review Assessment: Checking Correctness Of Derivations And Theory:**

I assessed the sensibility of the derivations and theory.

**Review Assessment: Checking Correctness Of Experiments:**

I assessed the sensibility of the experiments.

**Review Assessment: Thoroughness In Paper Reading:**

I read the paper at least twice and used my best judgement in assessing the paper.

---

> ### Author Response · Authors · 2019-11-11
> **Author Response to Official Blind Review #3**
>
> Thank you for your helpful feedback, and we certainly appreciate your time evaluating our work! Below we provide our specific answers to your comments.
>
> 1) Since $D^{'^{f}}_k \subset D^f_k$, would the model minimize w.r.t. both the loss of $L(h(x, \theta), 0)$ and $L(h(x, \theta), 1)$? Would this cause unstable training?
>
> In practice we use the objective in Equation (5), which does not involve $f$ to train detectors. In other words, we treat the classifier $f$ to be fixed, and therefore we only train the detectors $h_k$s. Regarding the overlap of sets, we first clarify that $D^{'^{f}}_k$ is the set of adversarially attacked samples from other classes that fool the classifier network $f$ to think the samples belong to class $k$. And, $D^f_k$ is the set of un-attacked/clean samples that the classifier, $f$, identifies (correctly or incorrectly) as class $k$. In theory, it could happen that the set $\mathcal{D'}_{\backslash k}=\{x_i+\delta_i: x_i\in\mathcal{D}_{\backslash k}\}$ and $\mathcal{D}_k$ have common elements. As an example, consider two data points  $x_i\in \mathcal{D'}_{\backslash k}$ and  $x_j\in\mathcal{D}_k$, and consider the unlikely case that $x_i$ would be transformed to $x_j$ using the perturbation $\delta=x_j-x_i$, if $\delta\in \mathcal{S}$. Now the binary cross-entropy loss for these two points,i.e., $x_j$ and $x_i+\delta$, are $\text{BCE}(h(x_i+\delta), 0) + \text{BCE}(h(x_j), 1) =  - \log(1-h(x_j))-\log(h(x_j))$ (assume $h(x_j)$ to be the sigmoid output), where we used the fact that $x_i+\delta=x_j$. This is in fact a convex function of $h(x_j)$ and has the global minimum at point $h(x_j)=0.5$. But in practice this is highly unlikely to happen, especially when we are using numerical optimization and have constraints on the perturbation limit.
>
> 2) Maybe I missed it, but it seems that the objective function in Eq. (5) is based on the adversarial detector. How could the classification performance of classifier $f$ be guaranteed in training?
>
> Following our previous comment, in practice, we use the objective in Eq. (5), which treats the classifier $f$ to be unchanging. To further elaborate on this, we first train the classifier $f$ and then partition the training data using $f$'s outputs. Now assuming that $f$ is complex enough to fit the training set reliably, the partition based on $f$ would be equivalent to (or very close to)  the partition based on the ground-truth labels. Therefore, we can omit $f$ from objective (4) and use the alternative objective (5) to train our detectors. In this case, the training of detectors is entirely independent of the classifier, and the classification performance of the classifier won't be affected.
>
> 3) What does the cross mark mean in Figures 2(b) and 4(b)?
>
> The cross marks indicate the performances of the "robust classifier" trained with standard adversarial training [1] (in Section 4.3 we provide a discussion about the performance metric). In Fig. 2(b) the cross mark indicates the performances of a robust classifier under eps=0.3 and eps=0.4 perturbations on the MNIST dataset, and in Fig. 4(b) the performances under eps=8 and eps=12 perturbations on the CIFAR10 dataset. For both classifiers, we used the pre-trained models provided by Madry et al. [1] to compute the performances.
>
> [1] Madry, Aleksander, et al. "Towards deep learning models resistant to adversarial attacks." arXiv preprint arXiv:1706.06083 (2017).

---

### Public Comment · ~Anthony_Wittmer1 · 2019-10-18
**About the evaluation on the black-box attack**

 Hi, thanks for this contribution.

It seems that the evaluation on the black-box attack is missing in this paper, which is important, because that if a model causes obfuscated gradients, black-box attacks perform better than white-box attacks[1].

[1] Obfuscated Gradients Give a False Sense of Security: Circumventing Defenses to Adversarial Examples. ICML 2018

---

> ### Author Response · Authors · 2019-10-23
> **Nattack result**
>
> Hi Anthony,
>
> Thanks for the suggestion. Black-box testing is indeed missing in our experiments. Here we report the black-box test results using the CIFAR10 class 0 base detector. We used PGD attack and Nattack to optimize (maximize) the logit output of the base detector. We used 100 test samples of classes from 1 to 9 to run the test, and the average logit outputs achieved by PGD and Nattack are respectively -17.53 and -17.73 (higher is better). In particular, Nattack outperforms PGD in 11 out of the 100 test cases. So in most cases PGD performs better, which indicates that the proposed objective didn't cause obfuscated gradients. Speed-wise, the 1000-iteration based Nattack runs about 150 times slower than 100-iteration based PGD attack. The script for reproducing these results is at  https://github.com/nhLKeO/AAT-CIFAR10/blob/master/eval_base_detector_Nattack.py . P.S. we  updated some image generation results on https://github.com/nhLKeO/AAT-CIFAR10 .

---

### Decision · Program_Chairs · 2019-12-19

**Decision:**

Accept (Poster)

**Comment:**

This work addresses the problem of detecting an adversarial attack. This is a challenging problem as the detection mechanism itself is also vulnerable to attack. The paper proposes asymmetrical adversarial training as a robust solution. This approach partitions the feature space according to the output of the robust classifier and trains an adversarial example detector per partition. The paper demonstrates improvements over state-of-the-art detection techniques.

All three reviewers recommend acceptance of this work. Some positive points include the paper being well-written with strong experimental evidence. One potential difficulty with the proposed approach is the additional computational cost associated with a per class adversarial attack detector. The authors have responded to this concern by claiming that the straightforward version of their approach is K times slower (10 in the case of 10 classes), but their integrated version is 2x slower as they only run the detector associated with the example-specific class prediction. We encourage the authors to include a discussion on computational cost in the final version. In addition, there was a community comment about black-box testing which will be of relevance to many in the community. The authors have already provided additional experiments to address this question as well as code to reproduce the new experiment.

Overall, the paper addresses an important problem with a two-step solution of training a robust model and detecting potentially perturbed samples per class. This is a novel solution with comprehensive experiments and therefore recommend acceptance.